
# Optimal probabilistic placement of facilities using a surrogate model for 3D tsunami simulations

Kenta Tozato[1], Shuji Moriguchi[2], Shinsuke Takase[3], Yu Otake[1], Michael R. Motley[4], Anawat Suppasri[2], and Kenjiro Terada[2]

[1]Department of Civil and Environmental Engineering, Tohoku University, Aza-Aoba 6-6-06, Aramaki, Aoba-ku, Sendai 980-8579, Japan
[2]International Research Institute of Disaster Science, Tohoku University, Aza-Aoba 468-1, Aramaki, Aoba-ku, Sendai 980-8572, Japan
[3]Department of Civil Engineering and Architecture, Hachinohe Institute of Technology, 88-1 Ohbiraki, Myo, Hachinohe, Aomori 031-8501, Japan
[4]Civil and Environmental Engineering, University of Washington, 201 More Hall, Box 352700, Seattle, Washington, 98195-2700, USA.

**Correspondence:** Kenta Tozato (kenta.tozato.t2@dc.tohoku.ac.jp)

**Abstract.** Tsunamis are associated with numerous uncertainties. Therefore, there has been an emphasis on setting the placement of infrastructure facilities based on probabilistic approaches. However, advanced numerical simulations have been often insufficiently utilized due to high computational costs. Therefore, in this study, we developed a framework that could efficiently utilize the information obtained from advanced numerical simulations for probabilistic assessment and investigation of

the optimal placement of facilities based on calculated probability. Proper orthogonal decomposition (POD) techniques were employed for utilizing the data from the numerical simulations for probabilistic risk evaluation. We constructed a surrogate model in which POD was efficiently used to extract the spatial modes. The results of the numerical simulation were expressed as a linear combination of the modes, and the POD coefficients were expressed as a function of the uncertainty parameters to represent a result of an arbitrary scenario at a low computational cost. We conducted numerical simulations of the 2011

tsunami off the Pacific Coast caused by Tohoku Earthquake as an example of the method proposed in this study. The tsunami reached the target area, and the fault parameters of "slip" and "rake" were selected as the target uncertainties. We then created several scenarios in which these parameters were changed and conducted further numerical simulations using POD to construct a surrogate model. We selected the maximum inundation depth in the target area and the maximum impact force that acts on the buildings as the target risk indices, and we constructed a surrogate model of the spatial distributions of each indicator. Fur-

thermore, we conducted Monte Carlo simulations using the constructed surrogate model and the information on fluctuations in uncertainties to calculate the spatial distribution of the failure criterion exceedance probabilities. We then used the Monte Carlo simulation results and a genetic algorithm to identify the optimal placement of facilities based on probability. We also discuss how the optimal placement changes according to differences in risk indices and the differences between parallel and series systems. The failure scenarios for each system are also discussed based on the failure probability. We show that the

proposed method of efficiently utilizing advanced numerical simulation information was useful for conducting probabilistic hazard assessments and investigating the optimal placement of facilities based on probability theory.



## 1 Introduction

Numerical analysis techniques for tsunamis have developed considerably over the years, and high-accuracy hazard assessments and predictions have now become possible (Qin et al., 2018; Xiong et al., 2019). Natural disasters such as tsunamis have numer-

ous uncertainties; hence, conducting probabilistic risk assessments that consider such factors is important. However, advanced numerical analyses are generally computationally expensive. Furthermore, they are not very compatible with probabilistic assessments and require multiple trials. Therefore, a framework that can efficiently combine these aspects is required.

Probabilistic risk assessments of disasters have been researched for many years in the seismology fields. Among these, the study by Cornell (Cornell, 1968) is considered groundbreaking. Many research results have been reported on probabilistic seis-

mic hazard analyses (PSHA) (e.g., McGuire, 1977; Ishikawa and Kameda, 1988). Furthermore, probabilistic tsunami hazard analyses (PTHA) have been established based on PSHAs; they have been proposed as a method for understanding the relationship between tsunami heights and exceedance probabilities in a specified period. PTHAs are practical methods for mitigating disaster damage, and many studies on PTHA have been reported (e.g., Annaka et al., 2007; Power et al., 2007; Thio et al., 2007; Mitsoudis et al., 2012; Fukutani et al., 2021, 2015; Park and Cox, 2016); many review papers have also reviewed the

topic (Geist and Parsons, 2006; Mori et al., 2017).

Furthermore, the appropriate placement of infrastructure and evacuation facilities is important for minimizing damage due to tsunamis. Numerous studies have used a probabilistic approach for the optimal placement of network systems and facilities by considering the uncertainties of disasters such as earthquakes or tsunamis. Such research examples include risk assessments for infrastructure system networks (Gomez and Baker, 2019; Miller and Baker, 2015), optimization of relief supply bases and

their delivery (Cavdur et al., 2020a, b; Maharjan and Hanapla, 2020), optimal placement of public and evacuation facilities (Park et al., 2012; Zhang and Yun, 2019; Doerner et al., 2008), emergency medical service networks (Mohamadi and Yaghoubi, 2017), and optimal allocation of budgets for disaster countermeasures (Rawls and Turnquist, 2010).

Thus, many research examples on the probabilistic hazard assessments of tsunamis and probabilistic optimal placement of facilities exist; however, few studies have sufficiently used the information obtained from advanced numerical analysis. In this

study, we applied the theory of mode decomposition using proper orthogonal decomposition (POD) (Kerschen et al., 2005) for solving such issues. The objectives of POD include extracting data characteristics or reducing data dimensions. POD is often treated as an equivalent of Karhunen–Loeve decomposition developed by Karhunen (1947) or Kosambi (1943); or principal component analysis (PCA) (Jolliffe and Cadima, 2016) developed by Hotelling (1933). POD has numerous application examples in a wide range of fields; furthermore, there are various application examples in the fields of seismic engineering

and tsunami engineering. For example, Ha et al. (2008) applied POD for reducing computational costs to construct a tsunami surrogate model. LeVeque et al. (2016) and Melgar et al. (2016) applied the Karhunen–Loeve expansion to consider the distribution of fault slips under various scenarios. Furthermore, Nojima et al. (2018) conducted research on the combination of model decomposition based on singular value decomposition and numerical simulations to predict the distribution of strong motion; Bamer and Bucher (2012) used nonlinear finite element methods to construct a surrogate model, using POD for the

prediction of the behavior of buildings. Moreover, Fukutani et al. (2021) constructed a surrogate model that used POD to im-



plement probabilistic inundation assessments. POD can extract the features of spatial and temporal distributions of risk indices; thus, it is suitable for the construction of surrogate models for disaster hazard assessment.

Given this context, in this study, we constructed a surrogate model by applying POD to advanced three-dimensional (3D) tsunami simulation results; we propose a method that uses this surrogate model to efficiently investigate the optimal placement of facilities such as infrastructure facilities, relief supply bases, and evacuation shelters based on probability theory. Research examples of probabilistic assessments of tsunami hazards using surrogate models of numerical simulations include the previously mentioned approach of using mode decomposition by Fukutani et al. (2021) and the approach of using response surfaces by Kotani et al. (2020)); however, no research examples exist in which the optimal placement of facilities has been investigated based on probability theory using a surrogate model. Therefore, in this study, we constructed a surrogate model using mode decomposition on information obtained from advanced numerical simulations. Moreover, we proposed a method that could efficiently investigate the optimal placement of facilities based on probability theory using this surrogate model.

The structure of this paper is as follows. In Section 2, the framework and methods used in this study are described. In Section 3, we applied the proposed method to the 2011 Off the Coast of Tohoku Earthquake as an application example that considers the actual damage and verified the validity of the constructed surrogate model; we also implemented Monte Carlo simulations to conduct probabilistic risk assessments. Furthermore, the Monte Carlo simulation results and a genetic algorithm were used to investigate the optimal placement of facilities, and the usefulness of the proposed method was discussed. Finally, Section 4 describes the conclusions.

## 2 Search method for the probabilistic optimal placement using a surrogate model

The proposed method is described in this section. In this study, a combined two-dimensional (2D) and 3D tsunami analysis was first conducted for multiple scenarios with different fault parameters. The maximum tsunami fluid force and the maximum inundation depth were adopted as the assessment indicators. Next, POD was applied to the results obtained from these analyses to extract the spatial modes of the tsunami fluid force and inundation depth; these spatial modes were used to construct a surrogate model; based on this model, the numerical analysis results of an arbitrary scenario could be calculated within a short time. Furthermore, this surrogate model and uncertainty parameter fluctuation information were combined to implement a Monte Carlo simulation and compute the probability distribution of the tsunami fluid force and inundation depth at each assessment point; a threshold value was set to create an exceedance probability map. Finally, the Monte Carlo simulation results were used, and a genetic algorithm was applied to investigate the optimal facility placement. Fig. 1 shows a flowchart of this study. From the next sub-section, the methods used in each portion are detailed. Of these, for the methods described in Sections 2.1 and 2.2, the methods proposed by Tozato et al. (2022) were adopted.

### 2.1 Tsunami simulation method

Numerical analyses were first conducted to construct a surrogate model of each indicator in the target region. A numerical analysis that combined 2D and 3D analysis was conducted; first, a wide-area 2D tsunami analysis was conducted, and the time


Tsunami Simulation (Section 2.1)

2D analysis ⟶ 3D analysis
Nonlinear shallow water equations          Navier-Stokes equation

Connection between 2D and 3D analysis

Mode Decomposition Based Surrogate Model (Section 2.2)

Spatial distribution for arbitrary uncertainty parameters

$$= f_1(\boldsymbol{\beta}) \quad\text{Mode 1} \quad + f_2(\boldsymbol{\beta}) \quad\text{Mode 2} \quad + \cdots + f_r(\boldsymbol{\beta}) \quad\text{Mode } r$$

(β: uncertainty parameters)

Monte Carlo Simulation    Probability density of tsunami force and inundation at each point
Maps of exceedance probability

Optimal Arrangement with Exceedance Probability (Section 2.3)

Optimization by Genetic Algorithm

| Initialization | Selection | Crossover | Mutation |
|---|---|---|---|
| | Fitness | | |
| | 0.9 | | |
| | 0.7 | | |
| | 0.5 | | |
| | 0.8 | | |
| | 0.3 | | |

**Figure 1.** Flowchart of probabilistic tsunami hazard analysis using the mode-decomposition–based surrogate model.





history of the tsunami wave height and flow velocity that were observed offshore of the target area were acquired. In this study, analyses were conducted using TUNAMI-N2 (Imamura, 1995; Goto et al., 1997). The following continuity and nonlinear long
wave equations are solved in the 2D analysis:

$$\frac{\partial \eta}{\partial t} + \frac{\partial M}{\partial x} + \frac{\partial N}{\partial y} = 0 \tag{1}$$

$$\frac{\partial M}{\partial t} + \frac{\partial}{\partial x}\left[\frac{M^2}{D}\right] + \frac{\partial}{\partial y}\left[\frac{MN}{D}\right] + gD\frac{\partial \eta}{\partial x} + \frac{gn^2}{D^{\frac{7}{3}}}N\sqrt{M^2 + N^2} = 0 \tag{2}$$

$$\frac{\partial N}{\partial t} + \frac{\partial}{\partial x}\left[\frac{MN}{D}\right] + \frac{\partial}{\partial y}\left[\frac{N^2}{D}\right] + gD\frac{\partial \eta}{\partial y} + \frac{gn^2}{D^{\frac{7}{3}}}N\sqrt{M^2 + N^2} = 0 \tag{3}$$

where $M$ and $N$ are the flow rates in the $x$ and $y$ directions, respectively, $\eta$ is the water level, $D$ is the total water depth, $g$ is gravitational acceleration and $n$ is the Manning roughness coefficient.

The obtained tsunami wave height and flow velocity were set as the boundary conditions, and the tsunami reaching the target area was analyzed. The time-series data of the wave height and flow velocity obtained from the 2D wide-area analysis
are stored and transferred to the 3D numerical analysis by linear interpolation in space. The interpolated values are given to the 3D analysis as input data.

A 3D analysis was performed in this portion to assess the fluid force acting on the buildings in the target area. We employed the following set of 3D Navier-Stokes and continuity equations in the analysis domain $\Omega_{ns} \in R^3$

$$\rho\left(\frac{\partial \boldsymbol{u}}{\partial t} + \boldsymbol{u} \cdot \nabla \boldsymbol{u} - \boldsymbol{f}\right) - \nabla \cdot \boldsymbol{\sigma} = 0 \tag{4}$$


$$\nabla \cdot \boldsymbol{u} = 0 \tag{5}$$

where $\rho$ is the mass density, $\boldsymbol{u}$ is the velocity vector, $\boldsymbol{\sigma}$ is the stress tensor, and $\boldsymbol{f}$ is the body force vector. Also, assuming a Newtonian fluid, the stress is calculated as

$$\boldsymbol{\sigma} = -p\boldsymbol{I} + 2\mu\varepsilon(\boldsymbol{u}) \tag{6}$$

where $p$ is the pressure, $\mu$ is the coefficient of viscosity, and $\varepsilon(\boldsymbol{u})$ is the velocity gradient tensor defined as

$$\varepsilon(\boldsymbol{u}) = \frac{1}{2}\left(\nabla \boldsymbol{u} + (\nabla \boldsymbol{u})^T\right) \tag{7}$$

To solve the governing equations of the 3D simulation, the stabilized finite element method (SFEM) is used in this study. The method in Takase et al. (2016) was used for the boundary conditions of the 2D and 3D analyses. For the tsunami uncertainty, two fault parameters (described later) were adopted as the uncertainty parameters, and numerical analyses were conducted for
several scenarios in which these parameters were changed. Specific analysis area setting conditions are shown in Section 3.


## 2.2 Construction of the surrogate model using mode decomposition

Proper orthogonal decomposition (POD) was used to extract the spatial modes in this study. POD can efficiently express data and extract the basis representing data characteristics. To apply POD to the data obtained from numerical analysis, a data matrix was first defined. When data from a given scenario $i$ are set as a vector and defined as $\boldsymbol{x}_i$ (called a data vector), then the result

of the data vectors corresponding to the number of scenarios arranged in the column direction is defined as follows:

$$X = \begin{pmatrix} | & & | \\ \boldsymbol{x}_1 & \cdots & \boldsymbol{x}_N \\ | & & | \end{pmatrix} \tag{8}$$

Here, $N$ refers to the number of scenarios, and the data vector is defined as a vector with a total of $n$ elements. Furthermore, the vertical line in Eq. (8) was used to indicate that the data vector is a column vector. Using this matrix, the covariance matrix of the data was expressed in the form of $\boldsymbol{C} = \boldsymbol{X}\boldsymbol{X}^T$; the eigenvalues represent the variance, and the eigenvectors represent

the spatial mode (characteristic of the spatial distribution). In this study, we assumed that the eigenvalues were arranged in descending order from the first mode, and the eigenvalue and eigenvector corresponding to the $j$-th mode were expressed as $\lambda_j$ and $\boldsymbol{u}_j$, respectively. Furthermore, in POD, the contribution rate of each mode is often used as a criterion for determining the number of dimensions to be reduced. The contribution rate is an index that shows how much each mode explains the data, and the contribution rate of the $j$-th mode is expressed as follows, using the eigenvalues.

$$d_j = \frac{\lambda_j}{\sum_{k=1}^{n} \lambda_k} \tag{9}$$

Furthermore, singular value decomposition was used to express the data matrix as follows using the eigenvalue $\lambda_j$ and the eigenvector $\boldsymbol{u}_j$.

$$X = \boldsymbol{U}\boldsymbol{\Sigma}\boldsymbol{V}^T = \begin{pmatrix} | & & | \\ \boldsymbol{u}_1 & \cdots & \boldsymbol{u}_p \\ | & & | \end{pmatrix} \begin{pmatrix} \sqrt{\lambda_1} & & \\ & \ddots & \\ & & \sqrt{\lambda_p} \end{pmatrix} \begin{pmatrix} - & \boldsymbol{v}_1 & - \\ & \vdots & \\ - & \boldsymbol{v}_p & - \end{pmatrix} = \boldsymbol{U}\boldsymbol{A} \tag{10}$$

Here, $\boldsymbol{U}$ is a matrix in which the modes are arranged in the column direction, $\boldsymbol{\Sigma}$ is a matrix in which the square roots of the

eigenvalues are arranged in diagonal terms, $\boldsymbol{V}$ is a matrix in which the eigenvectors of $\boldsymbol{X}^T\boldsymbol{X}$ are arranged, $p$ is the number of eigenvalues that is greater than zero, and $\boldsymbol{A} = \boldsymbol{\Sigma}\boldsymbol{V}^T$ is a matrix in which the POD coefficients are arranged. The relationship of singular value decomposition for the result of one scenario is given as follows:

$$\boldsymbol{x}_i = \sum_{k=1}^{p} \alpha_{ik}\boldsymbol{u}_k = \alpha_{i1}\boldsymbol{u}_1 + \cdots + \alpha_{ip}\boldsymbol{u}_p \tag{11}$$

Here, $\alpha_{ik}$ shows the $k$-by-$i$ column component of the matrix in which the POD coefficients are arranged. When modes with

low explainability for the data are removed, and the data are expressed in an approximate manner, the number of modes $r$ to be reduced is determined from the contribution rate and other indicators and is expressed as follows as a linear sum excluding




the modes with a low contribution rate.

$$\boldsymbol{x}_i \simeq \sum_{k=1}^{r} \alpha_{ik} \boldsymbol{u}_k = \alpha_{i1} \boldsymbol{u}_1 + \cdots + \alpha_{ir} \boldsymbol{u}_r \tag{12}$$

However, notably, a reduction in dimensions will result in a loss of the information contained in the omitted modes. Here,
if this data $\boldsymbol{x}_i$ is the result of the uncertainty parameter $\boldsymbol{\beta}_i$, then the POD coefficient of any uncertainty parameter $\boldsymbol{\beta}$ can be expressed. Therefore, next, the POD coefficient $\alpha_{ik}$ is expressed as a function $f_k(\boldsymbol{\beta})(k=1,...,r)$ of the input parameter that expresses the uncertainty. The surrogate model can be expressed as follows by expressing this as a function of the uncertainty parameter for each corresponding mode.

$$\hat{\boldsymbol{x}}(\boldsymbol{\beta}) = \sum_{k=1}^{r} f_k(\boldsymbol{\beta}) \boldsymbol{u}_k \tag{13}$$

In this study, the radial basis functions (RBF) (Buhmann, 1990) were used as the interpolation functions. RBF interpolation can be used to handle cases where the analysis scenarios are not evenly arranged in the parameter space. The function $f_k(\boldsymbol{\beta})$ corresponding to mode $k$ can be expressed as follows:

$$f_k(\boldsymbol{\beta}) = \sum_{i=1}^{N} w_i \phi(\boldsymbol{\beta}, \boldsymbol{\beta}_i) = \sum_{i=1}^{N} w_i \exp(-\gamma||\boldsymbol{\beta} - \boldsymbol{\beta}_i||^2) \quad (k=1,...,r) \tag{14}$$

Here, $\boldsymbol{\beta}_i$ is the input parameter group for scenario $i$, $w_i$ is the weight, and $\phi(\boldsymbol{\beta}, \boldsymbol{\beta}_i)$ is the basis function; $\gamma$ is a parameter
that determines the smoothness of the function. The weight of RBF interpolation can be obtained in the following form by substituting the correspondence between the known input parameter $\boldsymbol{\beta}_i$ and the coefficient value $\alpha_{ik}$ that expresses the output result.

$$\begin{pmatrix} \alpha_{1k} \\ \vdots \\ \alpha_{Nk} \end{pmatrix} = \begin{pmatrix} \phi(\boldsymbol{\beta}_1, \boldsymbol{\beta}_1) & \cdots & \phi(\boldsymbol{\beta}_1, \boldsymbol{\beta}_N) \\ \vdots & & \vdots \\ \phi(\boldsymbol{\beta}_N, \boldsymbol{\beta}_1) & \cdots & \phi(\boldsymbol{\beta}_N, \boldsymbol{\beta}_N) \end{pmatrix} \begin{pmatrix} w_1 \\ \vdots \\ w_N \end{pmatrix} \quad (k=1,...,r) \tag{15}$$

They can be expressed in their respective bold forms as follows:

$$\boldsymbol{\alpha}_k = \boldsymbol{\Phi} \boldsymbol{w}_k \quad (k=1,...,r) \tag{16}$$

Here, $\boldsymbol{\alpha}_k$ is a vector in which the coefficients of the $k$-th mode are arranged, and $\boldsymbol{w}_k$ is a vector in which the weights of $f_k(\boldsymbol{\beta})$ are arranged. The function using the weight obtained in Eq. (15) is expressed as an interpolation passing through all the referenced data points. However, cases in which the referenced data points change or vibrate at a local level can result in an interpolation wherein the accuracy of the physical meaning expression is low. To resolve such issues, we introduced a
regularization term when computing the weights. Specifically, we introduced L2 regularization called ridge regression (Hoerl and Kennard, 1970), and weight $\boldsymbol{w}_k$ was obtained by solving the following optimization problem.

$$\underset{\boldsymbol{w}_k}{\arg \min}(||\boldsymbol{\alpha}_k - \boldsymbol{\Phi} \boldsymbol{w}_k||_2^2 + \lambda||\boldsymbol{w}_k||_2^2) \tag{17}$$





This process is generally used in the field of machine learning to prevent overfitting; $\lambda$ indicates the degree of regularization. Introducing the regularization term allows for the prevention of local vibrations and enables a smooth interpolation. However,

care must be taken for cases in which regularization is introduced because this may not pass through all data points. Furthermore, the accuracy of interpolation depends on the RBF parameter $\gamma$ and regularization parameter $\lambda$; hence, these need to be appropriately determined. In this study, a combination of these parameters was determined by cross-validation (Stone, 1947) and Bayesian optimization (Močkus, 1975).

### 2.3    Search for the optimal placement using a genetic algorithm

A genetic algorithm (Holland, 1992) was used in this study for investigating optimal placement. Genetic algorithms search for approximate solutions of data, where multiple individuals whose solution candidates are expressed with genes are prepared, individuals with high fitness are preferentially selected, and solutions are searched for by repeating operations such as crossover and mutation. The problem targeted in this study includes an extremely large number of assessment points; furthermore, checking all combinations is extremely inefficient. Hence, we adopted an efficient genetic algorithm for the combination

optimization problem.

Figure 2 shows an overview of the genetic algorithm. In this study, the point number was placed in the component of each individual, and optimization of those combinations was performed with a genetic algorithm. First, the number of individuals was determined, and a combination of points to be selected was randomly determined for the initial individuals. Next, the fitness was calculated for the generated individuals. Two individuals that are to be the parents of the next generation were

then selected according to the obtained fitness. The parent selection method involves selecting individuals with high fitness as parents; low-fitness individuals are thus eliminated. The next generation of individuals is generated by randomly exchanging each component for the two selected parents. The location of exchange and the number of exchanges are randomly determined. This is repeated until the number of individuals in the next generation reaches the initially set number. In this study, we adopted an elite conservation strategy as a method to avoid deterioration of fitness during generational change, with settings

such that some of the top individuals with high fitness could be passed on to the next generation as is. The final next-generation individuals were determined by mutating each component of each individual with a certain probability. In this process, the point number may be duplicated within one individual, and in such cases, the duplicated point is randomly re-selected. This process was repeated until the fitness converged, and an optimal point combination was determined.

### 3    Application to cases assuming an actual tsunami

The method described in the previous section was applied to a problem in which an actual tsunami was assumed. In this study, we conducted a series of numerical analyses that considered the uncertainty with the 2011 Off the Coast of Tohoku Earthquake as the target. We applied mode decomposition on these results to construct a surrogate model of the numerical analysis, and we implemented Monte Carlo simulations in order to investigate the optimal placement of facilities based on probability theory.




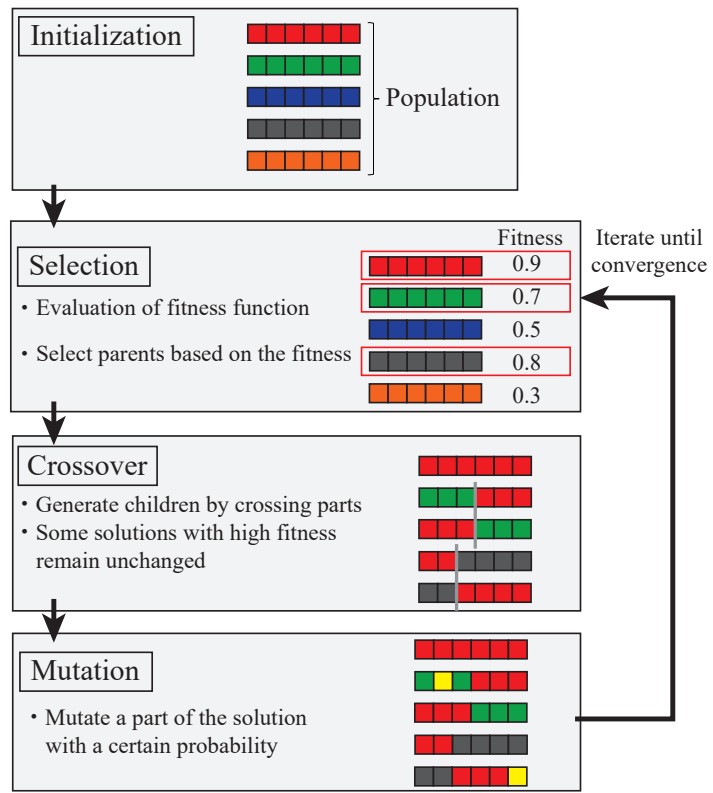

**Figure 2.** Flow of the genetic algorithm.

### 3.1 Tsunami simulation

We used the same numerical analysis results as those conducted in previous research (Tozato et al., 2022). The reader is referred

to the study by (Tozato et al., 2022) for details regarding the computational conditions. The parameter values at the time of the

Off the Coast of Tohoku Earthquake were set as mean values, with the slip varying between 0.7 and 1.4 times, and the rake

varying between $-20°$ to $+25°$. The illustration of the fault parameters is shown in Fig. 3. Table 1 shows all analysis cases.

In this study, the validity of the constructed surrogate model was verified by holding cases in which the rake was $\pm10°$ among

the 50 cases shown in Table 1; the remaining 40 cases were used to construct the surrogate model.

Figures 4 and 5 show the target area and snapshots of the analysis results of the inundation area for the mean case (S3R5),

respectively. Figure 6 shows the results of comparison of the inundation depth with the actually observed inundation depth. The

observed data were referenced from (The 2011 Tohoku Earthquake Tsunami Joint Survey Group, 2012), and the placement

of the observation points from A to H is shown in Fig. 4. These results show that although there was some deviation between

the numerical analysis results and observed results at locations far away from the shore, the observation values were generally

reproduced in the areas around the shore. The deviation between the numerical analysis results and the values observed in the


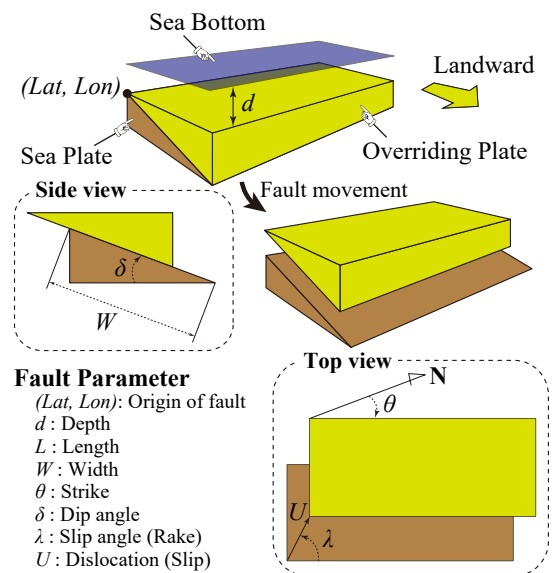

**Figure 3.** Illustration of the fault parameters. (adapted from Kotani et al. (2020)).

**Table 1.** Names of the calculation cases.

|  |  |  | Rake | | | | | | | | | |
|---|---|---|---|---|---|---|---|---|---|---|---|---|
|  |  | [°] | -20 | -15 | -10 | -5 | 0 | +5 | +10 | +15 | +20 | +25 |
|  | [%] | Normalized value | 0.753 | 0.815 | 0.877 | 0.938 | 1 | 1.062 | 1.123 | 1.185 | 1.247 | 1.309 |
|  | 70 | 0.7 | S1R1 | S1R2 | S1R3 | S1R4 | S1R5 | S1R6 | S1R7 | S1R8 | S1R9 | S1R10 |
|  | 85 | 0.85 | S2R1 | S2R2 | S2R3 | S2R4 | S2R5 | S2R6 | S2R7 | S2R8 | S2R9 | S2R10 |
| Slip | 100 | 1 | S3R1 | S3R2 | S3R3 | S3R4 | S3R5 | S3R6 | S3R7 | S3R8 | S3R9 | S3R10 |
|  | 120 | 1.2 | S4R1 | S4R2 | S4R3 | S4R4 | S4R5 | S4R6 | S4R7 | S4R8 | S4R9 | S4R10 |
|  | 140 | 1.4 | S5R1 | S5R2 | S5R3 | S5R4 | S5R5 | S5R6 | S5R7 | S5R8 | S5R9 | S5R10 |



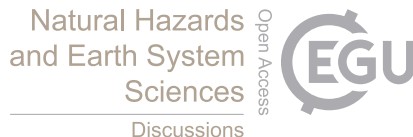

locations far away from the shore was thought to be because the outflows of the buildings were not considered; hence, the waves did not reach the locations far away from the shore.

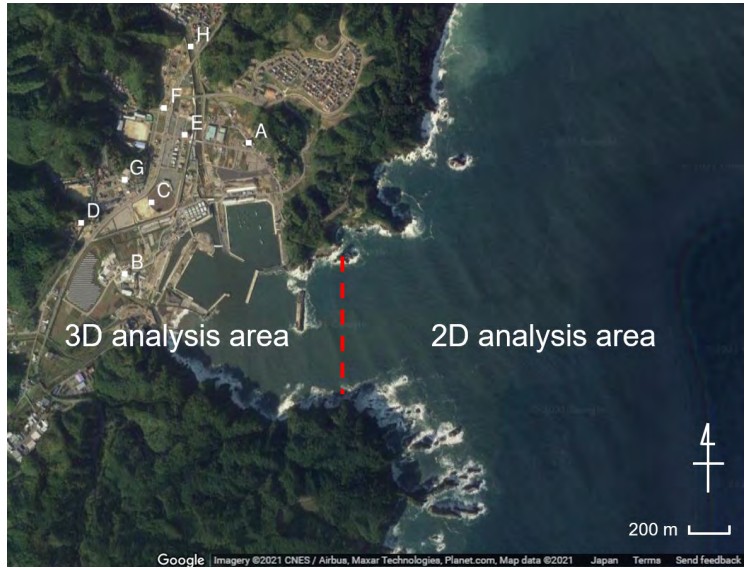

**Figure 4.** Boundary between 2D and 3D analyses areas. Points A to H are used to compare the inundation depths between the observational data and simulation results. (©Google Maps)

In this study, the physical values of the impact force acting on the buildings and the inundation depth were adopted as tsunami

risk indices. The effects of the tsunami fluid force have been considered even in recent design criteria (American Society of Civil Engineers, 2017; Nakano, 2017); therefore, we considered the fluid force acting on buildings as an additional risk index to the inundation depth. A 3D simulation was conducted to construct a surrogate model of the tsunami fluid force; however, the tsunami fluid force is strongly influenced by the direction that the building is facing, and it is difficult to assess the fluid force for each point. Therefore, the tsunami fluid force was assessed with a 2D grid size of approximately 10 m in this study. An

image of a mesh for evaluating the tsunami force is shown in Fig. 7. The target area was $2,145 \times 2,600$ m; hence, the number of assessment points in the POD was $n = 214 \times 260 = 55,640$.

### 3.2 Construction of a surrogate model with POD

POD was applied on the numerical analysis results to construct a surrogate model. When applying the POD, the data at each point were normalized in advance to a mean value of 0 and standard deviation of 1. Figure 8 shows the spatial modes extracted

by the POD from the first mode to the third mode. The values shown in the figure are values of the eigenvectors, and these were adjusted so that the maximum absolute value of the components in the eigenvectors was 1. The characteristics of the spatial distribution could be read for each physical quantity from the spatial modes. A comparison of the spatial modes of the maximum impact force and maximum inundation depth confirmed that the three modes shown here have similar characteristics.

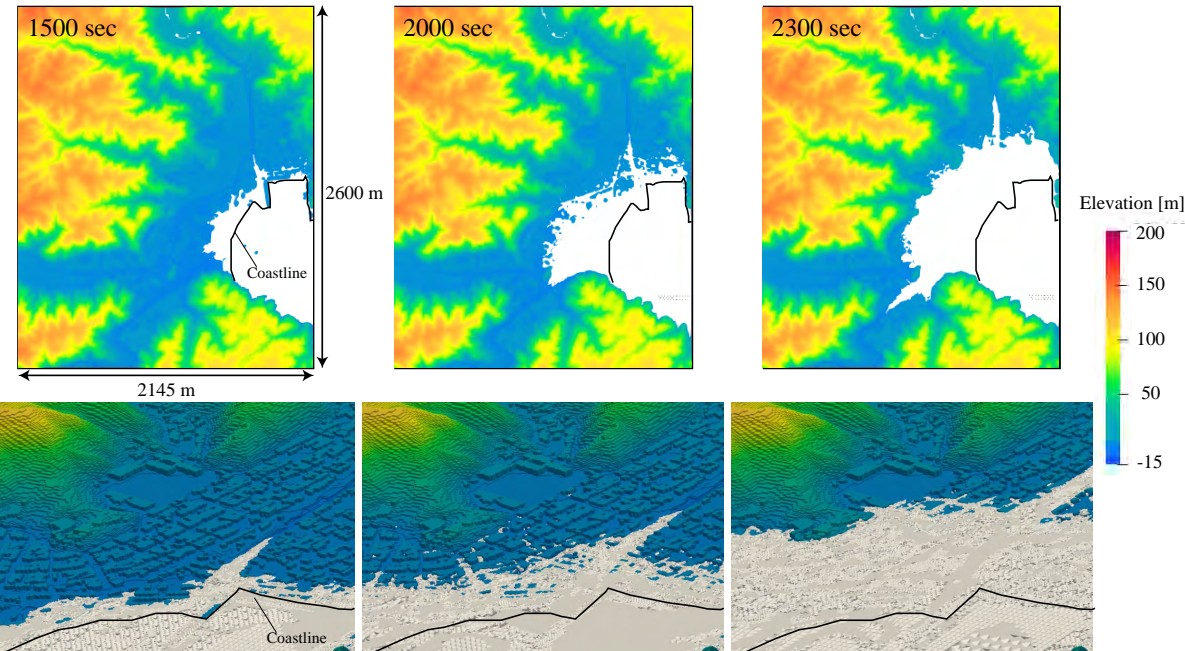

**Figure 5.** Snapshots of the tsunami runup obtained through 3D analysis.

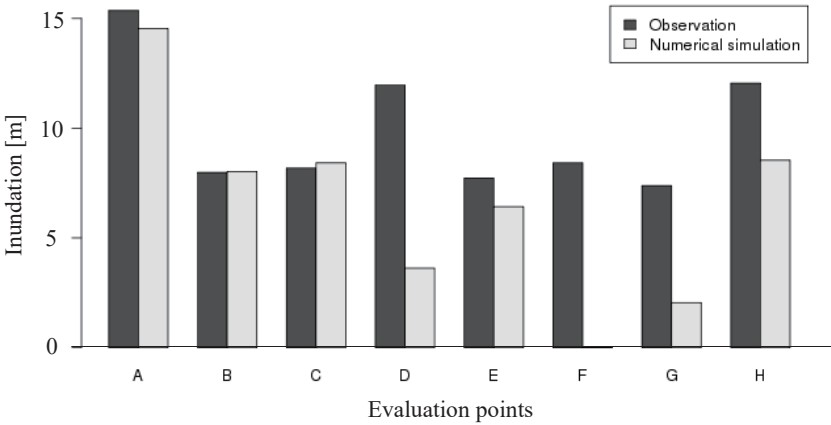

**Figure 6.** Comparison of the inundation between observational data and numerical simulation. (Observation data are provided by field survey results (The 2011 Tohoku Earthquake Tsunami Joint Survey Group, 2012)).



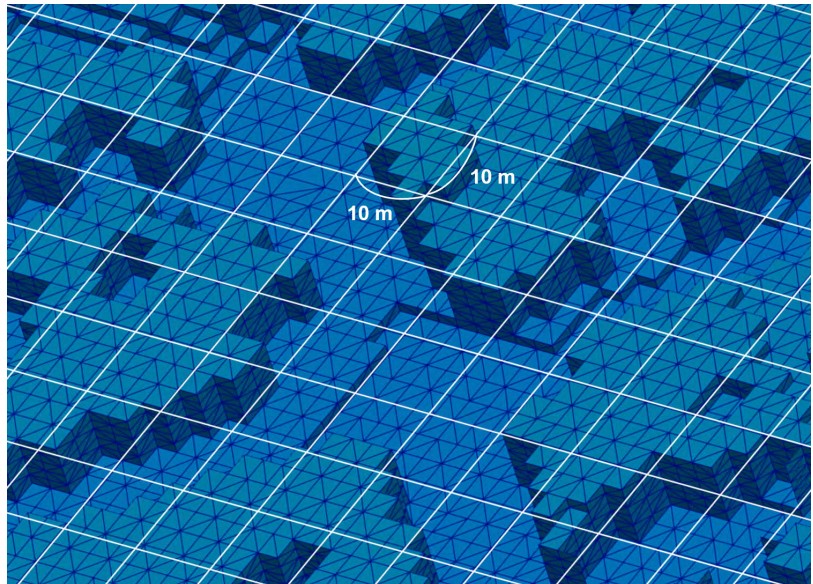

**Figure 7.** Image of a mesh for evaluating tsunami force.

For example, in the first mode, the sign was the same overall, and the value on the coast side was large; thus, the mode showed
an overall tendency where the coast side was the most affected by the tsunami, with the effect becoming smaller moving away
from the coast. In the second mode, a tendency could be seen where the maximum impact force and maximum inundation
depth were opposite at the east and west sides. In the third mode, a tendency could be seen where the maximum impact force
and maximum inundation depth were opposite at the north and south sides. These modes were thought to be related to the
inflow direction of the tsunami. Furthermore, higher-dimension modes included the characteristics of local sections. Figure 9
shows the contribution rates of the modes. The contribution rate of the first mode was extremely high for both the maximum
impact force and maximum inundation depth.

Next, the coefficients of each mode were expressed as a function of the uncertainty parameters. As previously mentioned,
RBF interpolation shown in Eq. (14) was used, and the regularization shown in Eq. (17) was introduced in the calculation to
obtain the weight. The accuracy of the surrogate model changed according to the RBF smoothness parameter and regularization
parameter; thus, it is important to appropriately determine these. In this study, these parameters were determined using cross-
validation.

The learning and verification cases used for cross-validation were obtained by taking a total of 40 cases used in the con-
struction of the surrogate model, removing four cases that correspond to the corners of the parameter space (S1R1, S1R10,
S5R1, S5R10), and dividing the remaining 36 cases between learning and verification cases for cross-validation. The corner
data were not used for the verification cases because these data were extrapolated. In this example, the number of divisions
between the learning and verification cases was set to 12. In other words, the policy was to construct a model with 37 cases


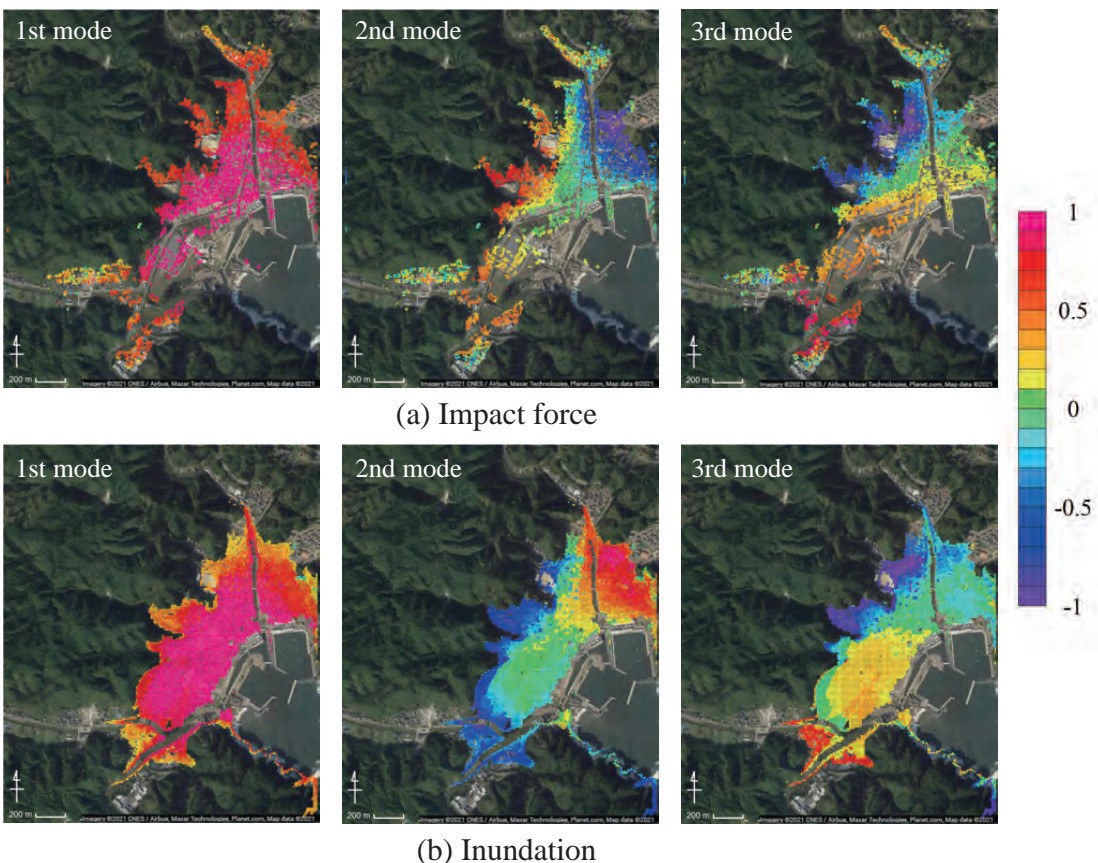

**Figure 8.** Spatial modes for each risk index extracted by POD. (left: 1st mode, middle: 2nd mode, right: 3rd mode) (©Google Maps)

for a single verification and conduct verification using three cases. Furthermore, the cross-validation error was calculated by comparing the reconstructed results using the spatial mode and taking the ratio of the mean absolute error to the mean value as shown in the following equation.

$$e_r = \frac{\frac{1}{nN} \sum_{i=1}^{n} \sum_{j=1}^{N} |x_{ij} - \hat{x}_{ij}|}{\frac{1}{nN} \sum_{i=1}^{n} \sum_{j=1}^{N} x_{ij}} \tag{18}$$


Here, $n$ is the number of assessed values, $N$ is the number of scenarios, and $x_{ij}$ is the numerical analysis result for scenario $i$ and point $j$. Furthermore, $\hat{x}_{ij}$ is the value for scenario $i$ and point $j$ when reconstructed with the surrogate model, and $e_r$ is the error when the number of modes is $r$.

Figure 10 shows the results of calculations of the maximum impact force and maximum inundation depth for the cross-

validation error in each mode. Cases in which no regularization term is present (Eq. (15)) are also shown. Bayesian optimization was used for the search in the parameter space, the number of searches was set to 80, and the upper confidence bound (UCB) strategy was used for the acquisition function. The state of error change confirmed that the accuracy of the surrogate model


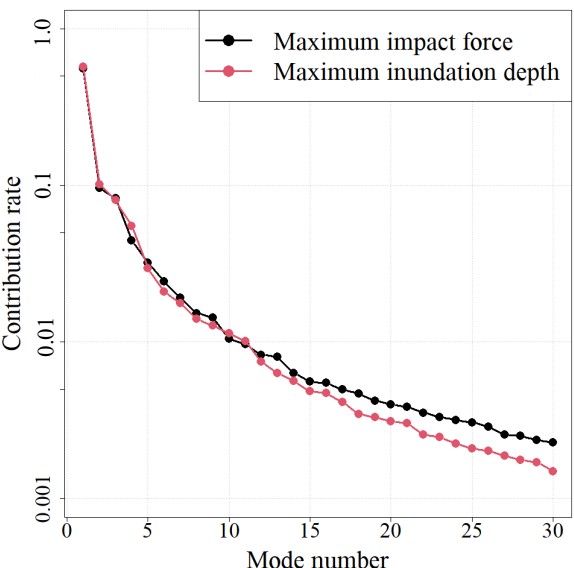

**Figure 9.** Contribution rates for each risk index.

was improved because the error value was reduced by introducing the regularization term. By introducing the regularization term, a robust surrogate model can be constructed.

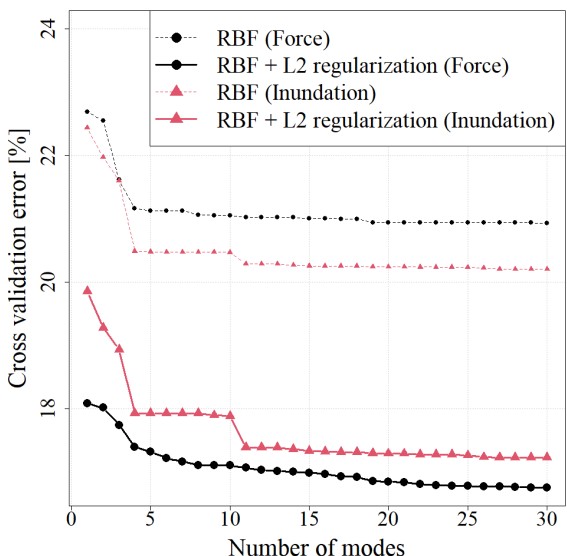

**Figure 10.** Cross-validation error for each mode.


Finally, the validity of the surrogate model was verified by comparing the results obtained from the numerical simulations

for the scenarios that were not used for the constructed surrogate model with the results of the constructed surrogate model.

Figure 11 shows a comparison of the numerical simulation results and the results obtained from the surrogate model for the

S3R3 scenario. Regarding the number of modes used in the surrogate model, the maximum impact force was set to 8 and

the maximum inundation depth was set to 11. The validity of the surrogate model can be confirmed because the figure shows

that the spatial distribution is generally reproduced. Furthermore, Fig. 12 shows the results of assessing the errors of the 10

scenarios that were held for verification. The error was calculated using Eq. (18). The error values as well that the numerical

analysis results were generally reproduced.

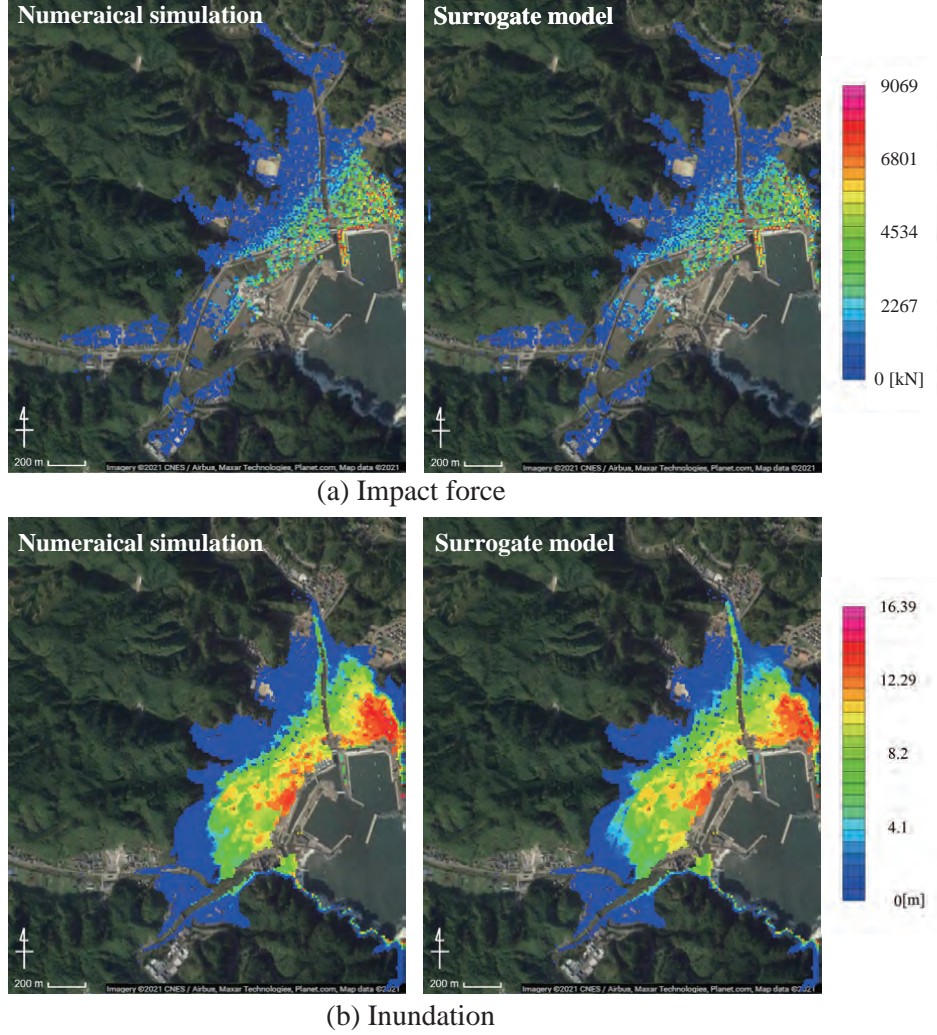

(a) Impact force

(b) Inundation

**Figure 11.** Comparison between results of numerical simulation and the surrogate model. (Scenario: S3R3) (© Google Maps)

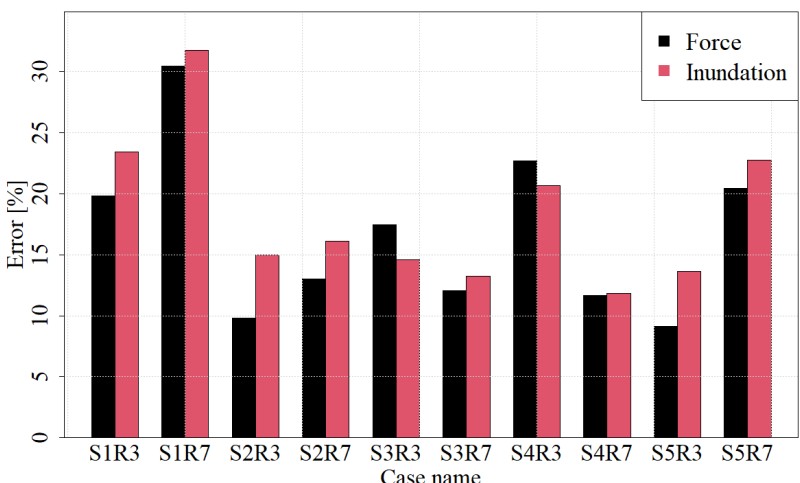

**Figure 12.** Errors between the numerical simulation and surrogate model for each validation scenario.

### 3.3 Monte Carlo simulation

We conducted probabilistic tsunami risk assessments using the surrogate model shown in the previous section. The surrogate
model enables the computation of a spatial distribution of risk indices at low computational cost; hence, many trials can be
secured at a relatively low computational cost, and a risk assessment that efficiently utilizes the advanced numerical simulation
results can be conducted.

In this study, we conducted a probabilistic assessment of the tsunami risk by applying Monte Carlo simulations. The un-
certainty of the changes in the input parameters must be quantitatively assessed in a probabilistic manner for Monte Carlo
simulations. We assumed that the slip and rake followed a normal distribution, and the probability distribution parameters were
set as shown in Table 2. The rationale for setting the values in this table is as follows. For the mean values, as mentioned in the
previous section, normalized values were used as the input parameters, so these were set as 1.0. Furthermore, for the standard
deviation of the slip, a value of 0.1 was used, which indicates a standard deviation value that is 10% of the mean value. For
a normal distribution, the spread of approximately three times the given standard deviation is present; thus, the value of three
times the standard deviation was set as this value so that the range of Table 1 in the previous section was covered. Furthermore,
for the standard deviation of the rake, Japan Society of Civil Engineering (2011) conducted probabilistic assessments where
the rake was varied by $\pm 10°$; therefore, in this study as well, we considered this degree of variation, and a value of 0.04 was
used, which was considered to result in a variation of approximately three times the standard deviation.

Monte Carlo simulations were conducted using the input parameters listed in Table 2 and the surrogate model. Specifically,
the uncertainty information of each input parameter was used to randomly generate a combination of the slip and rake; this
value was assigned to the function of the surrogate model coefficient component, and a value that was multiplied by the mode
was added according to the number of modes to calculate the spatial distribution of the risk index. This was repeated for the





**Table 2.** Information on the variation of uncertainty parameters.

| Parameter | Mean | Standard deviation |
|-----------|------|--------------------|
| Slip | 1.0 | 0.1 |
| Rake | 1.0 | 0.04 |

number of trials, which was set to 10,000 in this study, and the probability density distributions of the maximum impact force and maximum inundation depth at each point were calculated.

Maps of exceedance probability are obtained from the results of Monte Carlo simulations. The exceedance probability at each evaluation point is calculated assuming the failure that can be defined by the criteria of risk indices. Based on the previous studies (Suppasri et al., 2013, 2019), we defined the criteria as 176 kN for the maximum impact force and 3 m for the maximum inundation depth. The obtained exceedance probability maps for both risk indices are shown in Fig. 13. In both maps, there is a tendency that high exceedance probabilities arise near the coast and low exceedance probabilities occur farther away from the

coast. On the other hand, there are some differences between the maps locally. For example, the exceedance probability of the maximum impact force tends to be high in the areas where there are many buildings. Since the computational cost of surrogate models are quiet low, such probabilistic maps can be easily obtained.

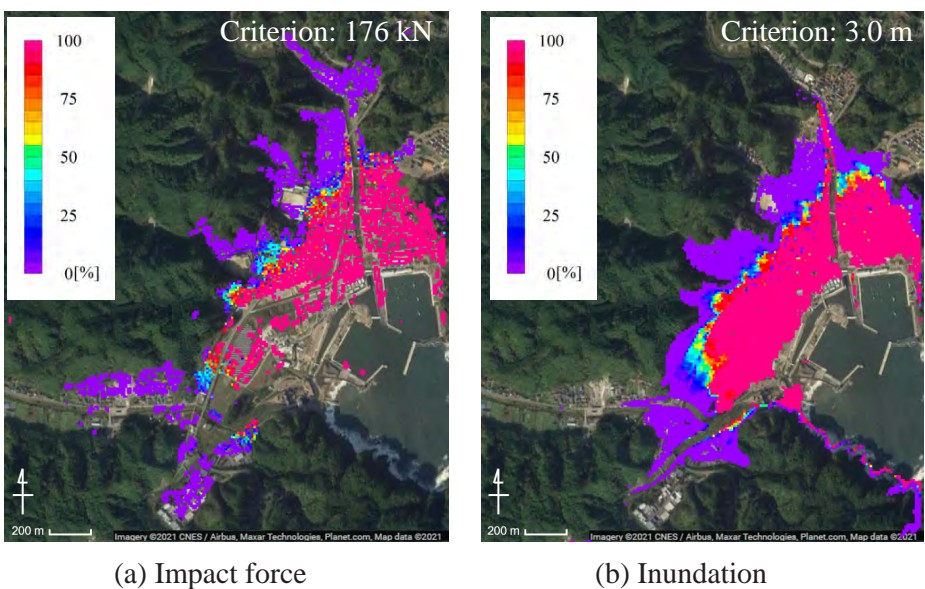

      (a) Impact force          (b) Inundation

**Figure 13.** Spatial distribution for the exceedance probability. (©Google Maps)

### 3.4 Optimal placement of system using genetic algorithms

The results of Monte Carlo simulation are used to investigate an optimal placement of the system. The optimal placement is examined based on the concepts of both parallel and series systems, and the results are compared. In the case of a parallel system, the system failure is considered to occur when all of its components have failed. On the other hand, regarding a series system, the system is considered to be have failed when any one of its components is destroyed.

In this study, the optimization problem is defined to select the components of the system from the evaluation points used in the construction of the surrogate model. The areas that have 25% or higher exceedance probabilities for each failure criteria are set as the target areas. The reason why the components of the system were selected only in these high risk areas is that a point with an exceedance probability of 0% should be always selected if there are such points in the target area, and this does not result in a optimal problem. This condition is not so realistic, but we employed it as a calculation condition to clearly represent the performance of the proposed method.

For the settings of the genetic algorithm, the number of individuals was set to 200, and the mutation probability was set to 10% for each component of each individual. Furthermore, we adopted an elite conservation strategy; in each generation, when there is no other individual that simultaneously improves both objective functions, the individuals of the current generation are passed on to the next generation as is. Moreover, the solutions near the optimal solution are likely to be selected for the crossover. In this study, the solution convergence was set as that when the solution individuals do not change over 2,000 steps. We investigated cases in which the number of components is 4 for each failure criterion and each system. Since the solution may depend on on the initial conditions, we hence conducted three trials under the same conditions for each case.

The results of optimal placements based on the concepts of the parallel and series systems for each risk index are shown in Fig. 14. In these figures, the results are also compared with the placements determined based on a simple strategy, in which the components of the system are selected in order of lower exceedance probability. White points shown in Fig. 14 indicate the components of the system selected by the genetic algorithm, and black points show the components selected in order of lower exceedance probability. System failure probabilities for each placement are shown in Table 3. The system failure probabilities are expressed as the probability of failure of all components for the parallel system and the probability of failure of one or more components for the series system.

**Table 3.** System failure probability for each risk index and each system.

| | Impact force | | Inundation | |
|---|---|---|---|---|
| | Parallel | Series | Parallel | Series |
| Minimum failure probability [%] | 20.94 | 29.62 | 22.50 | 28.03 |
| Genetic algorithm [%] | 19.40 | 28.09 | 0.10 | 25.50 |

According to the results shown in Fig. 14, it is found that the selected components are placed away from the coast line. This tendency comes from the fact that low failure probability points are generally located away from the coastal region. Different


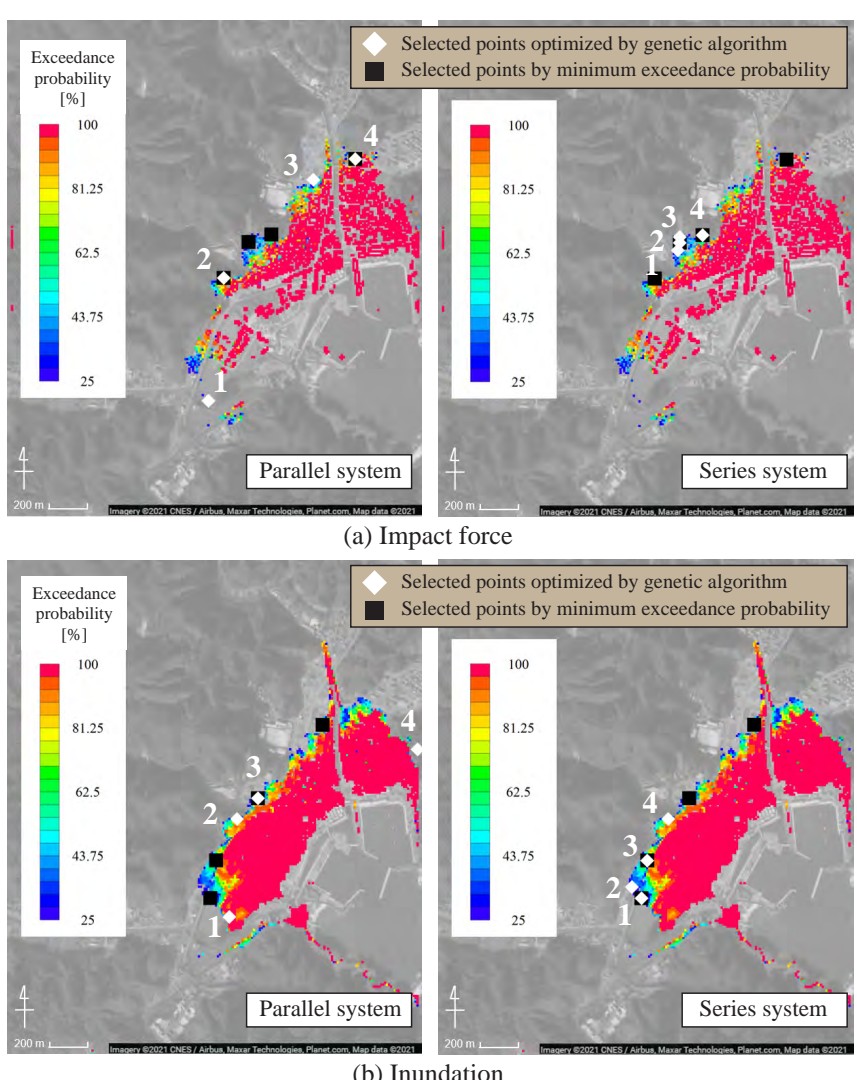

(a) Impact force

(b) Inundation

**Figure 14.** Optimal placements obtained from the genetic algorithm for each risk index. (©Google Maps) Selected points by minimum exceedance probability shows the results of placements determined in order of decreasing exceedance probability at each evaluation point. Only search areas are colored in this figure.





placements were obtained for the maximum impact force and inundation depth. Comparing the parallel and series systems, it can be seen that the components are selected by concentrating on similar locations in the series system, while the components are selected by spatially dispersing them in the parallel system. Regarding the series system, where all failures of one or more components would result in the destruction of the system, the components are concentrated in such an arrangement that the probability of exceedance is small. On the contrary, one or more of the components need only be safe in a parallel system. That

is why the components are spatially distributed.

Through the comparison of the obtained optimal placements for all systems, it can be seen that some common locations were selected for some components, but different locations were also selected for others. Although different placements were obtained, the risk for both parallel and series systems as shown in Table 3.

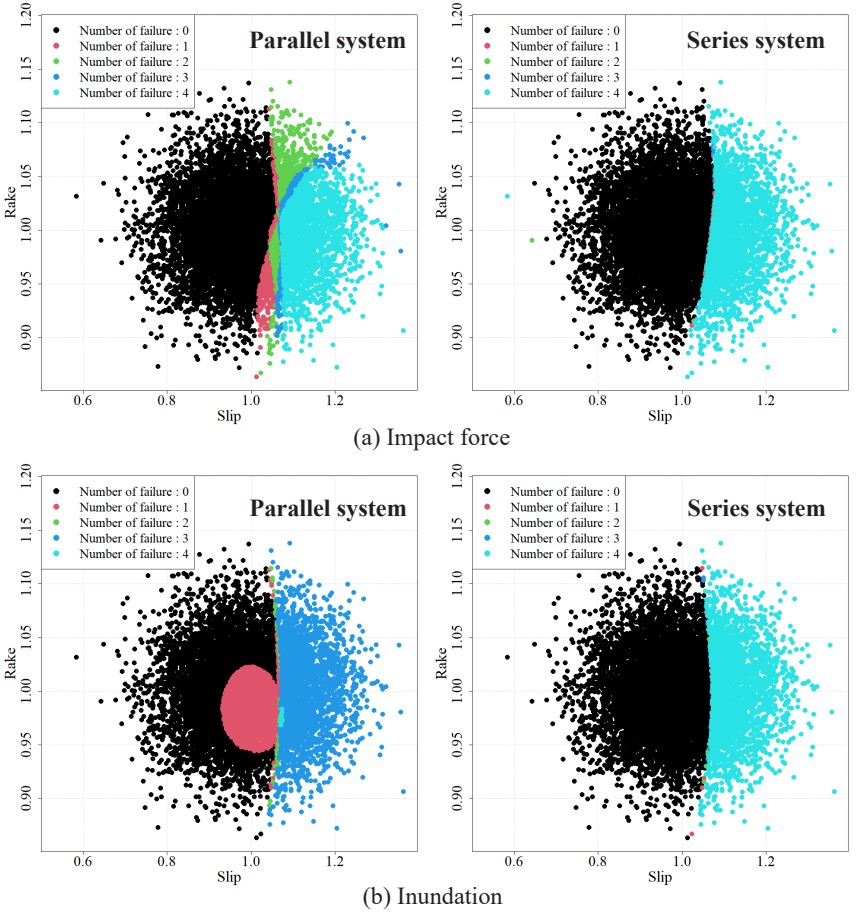

**Figure 15.** The scatter plot of the uncertainty parameters colored by the number of failure points for each placement.

A detailed analysis of the placements selected for each system was performed. Figure 15 shows the scatter plots of the

uncertainty parameters, slip and rake, considered in the Monte Carlo simulation, and the number of failure components is





colored. The placement for the parallel system optimizes around a smaller light blue area and the placement for the series system optimizes around a larger black area. According to Fig. 15, the slip mainly contributes to the failure of each point and the system for all placements because no failure occurs in smaller-slip cases and the failure occurs in the high-slip case. The rake contributes to the parallel system placement because the number of failure points changes in the same slip values for the

parallel system.

In the series system, the scenarios are separated into two situations; all safe and all destroyed. In contrast, the parallel system shows a more complex placement pattern. Figure 16 shows a colored scenario in the uncertainty parameter space. As can be seen in Fig. 16, regarding the maximum impact force, it can be seen that the components are selected such a way that they are not destroyed depending on the value of the rake. On the other hand, in the case of the inundation depth, it can be confirmed

that component 4 has unique tendency compared to the other components. As we can understand from this numerical example, the optimal placement is efficiently discussed by the method proposed in this study.

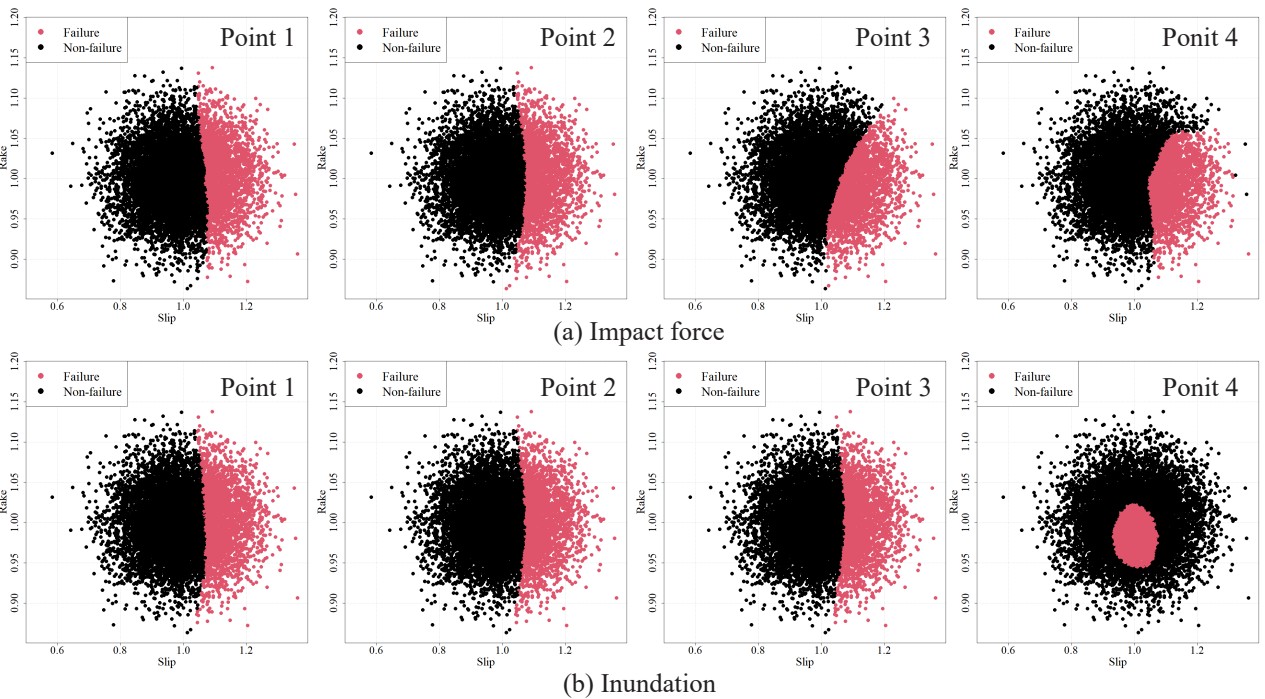

**Figure 16.** The scatter plot of the uncertainty parameters colored by failure or non-failure for each placement in inundation case.

Using the results of the Monte Carlo simulation using the surrogate models, the optimal system placements can be probabilistically investigated based on information from the advanced numerical simulation. This method could be used to solve the problem of optimal placement of facilities such as relief bases, shelters, and infrastructure facilities during disasters.



## 4   Conclusions

In this study, we used advanced 3D simulation results to construct a surrogate model, and we used this model to propose a method that enabled the efficient investigation of the optimal placement of facilities based on probability theory. We constructed a surrogate model by applying POD to 3D numerical simulation results of tsunamis, and Monte Carlo simulations that used this surrogate model were conducted to show that it was possible to assess the probability density distribution of risk indices at all points within the target area and the spatial distribution of exceedance probability with a low computational cost. Furthermore, we show that applying the genetic algorithm to Monte Carlo simulations enabled the search for the optimal placement of parallel and series systems, which minimized failure in a probabilistic manner. In this way, the proposed method enabled the investigation of the optimal placement of facilities in a probabilistic manner by efficiently utilizing the information obtained from advanced numerical simulations.

The uncertainty parameters in this study were limited to two when conducting assessments; however, there are more uncertainties than this in actual phenomena. Thus, it is important to conduct a probabilistic risk assessment that considers this fact. Furthermore, the extent to which each uncertainty parameter will fluctuate must be assessed in advance to determine cases in which numerical simulations or Monte Carlo simulations are to be conducted. The surrogate model that was constructed with the proposed method is capable of assessments with sufficient accuracy for interpolation (within the range of uncertainty parameters for which numerical simulations were conducted); however, the accuracy often decreases for cases of extrapolation (outside the range of uncertainty parameters for which the numerical simulations were conducted); thus, an assessment of uncertainty fluctuation is important. In addition, since the accuracy of the surrogate model changes according to the number of spatial modes, it's necessary to establish a way of properly determining the number of modes in future works. It is also noted that we calculated the exceedance probability with the destruction criterion as a constant when investigating the exceedance probability and optimal placement; however, the destruction criteria vary according to the building material. Thus, the use of this information would enable a more advanced probabilistic risk assessment and optimal placement of facilities.

## Appendix A:  Verification of the validity of the numerical analysis method

In this study, we conducted a comparison with the experimental results of the study by Winter et al. (2020) and verified the validity of the 3D analysis method adopted in this study. In the study by Winter et al. (2020), experiments were conducted on the fluid force acting on the structure while changing the structural placement; in this study, we conducted a comparison between the experiment results under the conditions shown in Fig. A1 among the aforementioned changes in structure placement with the analysis results. Details of the experiment are as shown in the study by Winter et al. (2020), and a comparison was performed with the numerical analysis results regarding the temporal changes in the water level on the front side of the structure and the fluid force acting on the structure. Fig. A2 shows the comparison results for each case. As shown in the figure, the validity of the adopted numerical analysis could be confirmed because the fluid force and water depth tendencies were generally captured. Furthermore, Fig. A3 shows a snapshot of the simulation results.




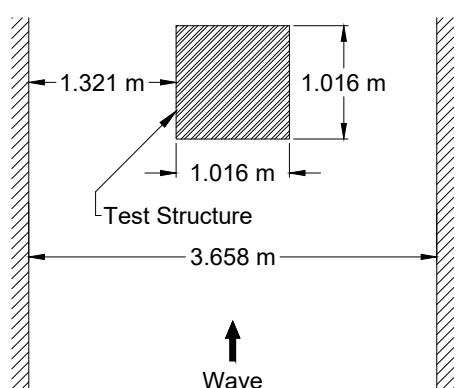

**Figure A1.** Configuration of the test structure.

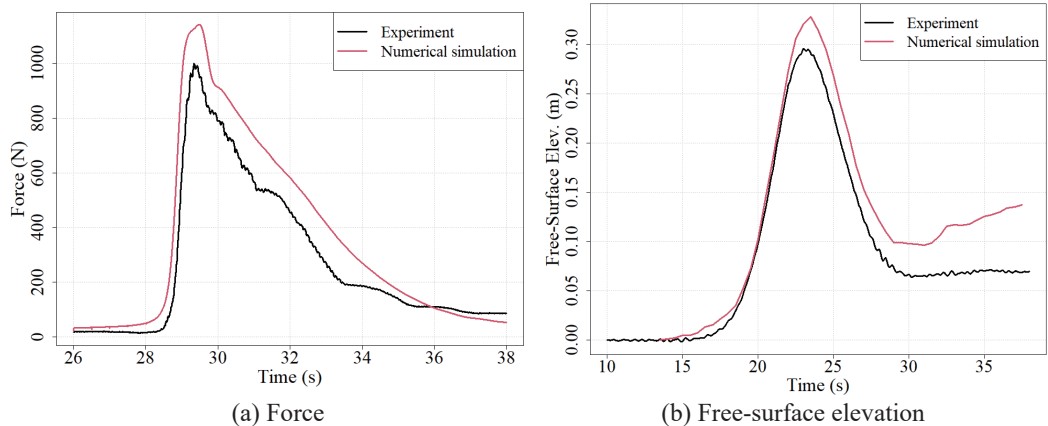

**Figure A2.** Comparison between the experimental results and numerical simulation results. ((a) Force, (b) Free-surface elevation)

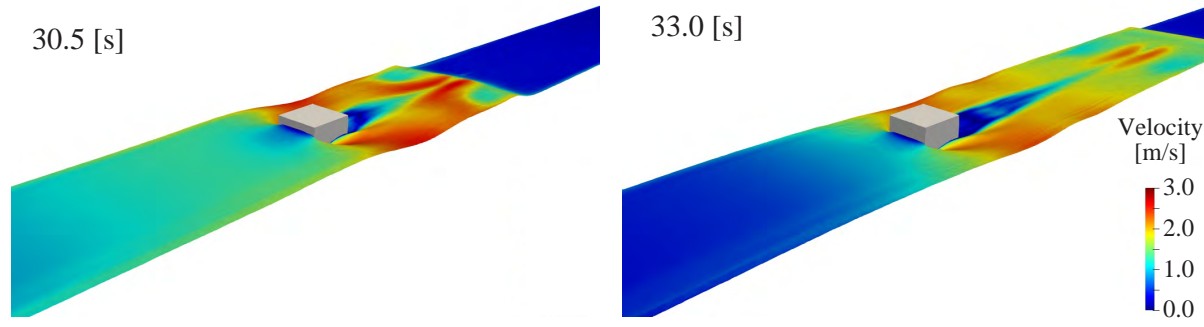

**Figure A3.** Snapshot of the numerical simulation result.



*Code and data availability.* Source code and details of the tsunami simulation were sourced from Kotani et al. (2020). Outputs of the simulations are available the Zenodo open-access repository at https://doi.org/10.5281/zenodo.6394294 (Tozato , 2022).

*Author contributions.* KTo contributed to methodology, analyses, and preparing the original manuscript. SM contributed to the conceptualization and methodology, and prepared the manuscript. ST contributed to the numerical simulations. YO and AS contributed to the methodology and conceptualization. MM contributed to the investigation and reviewing the manuscript. KTe contributed to the supervision and reviewing the manuscript.

*Competing interests.* The authors declare that they have no conflict of interest.



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
