# Peer review of "Optimal probabilistic placement of facilities using a surrogate model for 3D tsunami simulations"

_Natural Hazards and Earth System Sciences, 2022_

## Referee Comment (RC2)

**nhess-2022-208: Optimal probabilistic placement of facilities using a surrogate model for 3D tsunami simulations**

The manuscript presents a surrogate-based approach for probabilistic assessment and investigation of the optimal placement of facilities under tsunami forces. The topic is interesting and the manuscript is generally well written, and the presented approach could be used as an efficient alternative for probabilistic risk assessment of coastal infrastructure assets. Therefore, the manuscript is valuable to be published after some revisions.

1. In the numerical simulation of the tsunami, results obtained from the 2D analysis are used as the input of the 3D analysis. Please clarify the validity of this simplification and show some comparison results (if any) with theoretical results or experiments.

2. When calculating the tsunami force, what was the exact mesh sensitivity in the full-scale analyses? Did the authors conduct the mesh sensitivity to ensure that the mesh sensitivity is the same for the whole range of the investigated random parameters?

3. According to the results shown in Figure 12, most of the prediction errors are larger than 10%, indicating that the samples may not sufficiently large enough for the surrogate model to be generalized well. How to determine the appropriate number of samples is an important ingredient in data-driven approaches. I suggest the author should define (or add as the future research work) an appropriate stopping criterion to determine whether the sample size is sufficient or not, or resort to the so-called adaptive surrogate modelling to reach a balanced performance by sequentially adding new samples to the training set.

4. The manuscript is well written, yet the language should be double-checked to make it more readable, e.g., "Although different placements were obtained, the risk for both parallel and series systems as shown in Table 3" on Page 21.

---

## Author Comment (AC1)

**Response to referee comment 1**

Evaluation:

This is a good work and I think can be published. But the text is unclear in some places and misses important citations. It is important that authors clarify the items mentioned below, and add the suggested citations in order to make their work more international. My comments are given below. I look forward to reviewing the revised version. Best regards.

Thank you for your valuable comments. Our responses are summarized below.

(1)
Abstract:

Abstract is too long and is non-conclusive. Please try to add more quantitative results in your abstract. Also I suggest making your abstract shorter.

Thank you for your kind suggestion. We will add quantitative results in the abstract and make it shorter.

(2)
Figure is very similar to author team another article (https://doi.org/10.5194/nhess-22-1267-2022). Please try to make this figure different from your already-published article.

Since this paper uses same tsunami simulation results written in our previous paper (https://doi.org/10.5194/nhess-22-1267-2022), some figures are cited from the paper. Some figures do not have citation information, we will add the information.

(3)
L26: what is meant by "…are not very compatible with probabilistic…?" please clarify.

As you pointed out, this sentence is unclear. We will revise the sentence.

(4)

L33/34: For PTHA, to give more diversity to your citations, please add the following two good articles: "Gopinathan et al. 2021" and "Heidarzadeh & Kijko 2011".

Gopinathan, D., Hidarzadeh, M., Guillas, S. (2021). Probabilistic Quantification of tsunami current hazard using statistical emulation. Philosophical Transactions of the Royal Society A, 477, 20210180. https://doi.org/10.1098/rspa.2021.0180.

Hidarzadeh, M., Kijko, A. (2011).  A probabilistic tsunami hazard assessment for the Makran subduction zone at the northwestern Indian Ocean. Natural Hazards, 56 (3), 577-593. https://doi.org/10.1007/s11069-010-9574-x.

L68: For the 2011 event please add a reference for clarity. I recommend the article by Prof Tsuji (Tsuji et al., 2011), as below:

Tsuji Y, Satake K, Ishibe T, Kusumoto S, Harada T, Nishiyama A, Kim HY, Ueno T, Murotani S, Oki S, Sugimoto M, Tomari J, Heidarzadeh M, Watada S, Imai K, Choi BH, Yoon SB, Bae JS, Kim KO, Kim HW (2011) Field surveys of tsunami heights from the 2011 Off the Pacific Coast of Tohoku, Japan, earthquake. Bulletin of the Earthquake Research Institute University of Tokyo 86:29–279 (in Japanese with English abstract)

L72: I think it is useful to add a line regarding the importance of this study. I recommend something like this: "The recent 2022 Tonga tsunami, which made global impacts (Heidarzadeh et al. 2022), showed that tsunami is an important costal disaster and studies like this work are needed".

The reference is:

Heidarzadeh, M., Gusman, A., Ishibe, T., Sabeti, R., ŠepiÄ, J. (2022). Estimating the eruption-induced water displacement source of the 15 January 2022 Tonga volcanic tsunami from tsunami spectra and numerical modelling. Ocean Engineering, 261, 112165. https://doi.org/10.1016/j.oceaneng.2022.112165.

Thank you for your suggestion. We will add these references in the revised manuscript.

(5)

Figure 1: the color bar cannot be read. Please ensure they can be read easily and add a legend for them. Are they "Wave amplitude (m)?", add them.

Spatial distributions shown in the Fig. 1 are just images, and color bars shouldn't have been included. We will delete them from the figure.
* * *
(6)

L112: The 3D model is not clear? Please add a few references for that and explain more about it.
* * *
Because the explanation of the 3D model is written in detail in section 3 and the previous paper (Tozato et al. (2022)), we will add explanation to make this clear.
* * *
(7)

L113: What is this boundary condition? Explain more about it by adding the boundary equation and a few references.
* * *
As you pointed out, the explanation of the boundary condition is inadequate. We will add a specific description of the boundary condition.
* * *
(8)

L120: What are these data? Sea level? Velocity? Force?
* * *
In this study, the data indicate tsunami force and inundation depth. We will add explanation in the revised manuscript to make it clearer for readers.
* * *
(9)

L131: what is Lamda_k? what is d_j? every parameter needs to be defined as soon as they are used.
* * *
$d_j$ represents the contribution rate for mode $j$ and $\lambda_k$ represents the $k$-th eigenvalue. Although the explanation of $\lambda_k$ is written at Lines 126-127 the explanation of $d_j$ is not written in the manuscript. We will add the definition of $d_j$.

(10)

L139: again some parameters are not defined. Please ensure you define al parameters as soon as they are used throughout the text.

Since it is unclear that $\alpha_{ik}$ is a component of matrix $\boldsymbol{A}$, we will add this explanation.

(11)

L200-204: it is not clear why authors considered only slip and rake for uncertainty? Please clarify this and try to convince the readers? Why not depth while depth is a very important factor regarding tsunami energy? Please clarify.

As you pointed out, there are some other important uncertainty parameters, such as depth. However, since this study mainly aims to propose a framework for the probabilistic optimal facilities placement, we employed the simple calculation condition that was considered in our previous work (Tozato et al. (2022)). We ask for your understanding.

(12)

L203: regarding rake, you could refer to the following good articles that studied rake variations through teleseimic inversions (Gusman et al., 2014; Heidarzadeh et al., 2017):

Gusman, A. R., Murotani, S., Satake, K., Heidarzadeh, M., Gunawan, E., Watada, S., & Schurr, B. (2015). Fault slip distribution of the 2014 Iquique, Chile, earthquake estimated from ocean-wide tsunami waveforms and GPS data. Geophysical Research Letters, 42, 1053-1060. https://doi.org/10.1002/2014GL062604.

Heidarzadeh, M., Murotani, S., Satake, K., Takagawa, T., Saito, T. (2017). Fault size and depth extent of the Ecuador earthquake (Mw 7.8) of 16 April 2016 from teleseismic and tsunami data. Geophysical Research Letters, 44 (5), 2211–2219. https://doi.org/10.1002/2017GL072545.

Thank you for the valuable information. We will add these references in the revised manuscript.

(13)

Figure 5: I assume that the elevation is "Topography elevation (m)". Please modify the legend.

Thank you for pointing it out. We will revise the legend.

(14)

L208: For the 2011 Tohoku data, add reference to Mori et al. (2011):

Mori, N., Takahashi, T., and The 2011 Tohoku Earthquake Tsunami Joint Survey Group, (2012), Nationwide post event survey and analysis of the 2011 Tohoku earthquake tsunami. Coastal Engineering Journal, 54 (1), 1-27. https://doi.org/10.1142/S0578563412500015.

Thank you for your kind suggestion. We will add this reference in the manuscript.

(15)

Figure 8: what is the color bar? Please write it in the figure next to the colour bar. This way the figure can be easier and readers can notice it instantly, instead of reading your caption.

The color bar in Figure 8 indicates the values of modes (eigenvectors) extracted by POD. The values are normalized so that the maximum absolute value is 1. We will add to the explanation in the manuscript.

(16)

L254: It is not clear how the maximum impact force is calculated? Please write the equation tat you used to calculate the impact force here. Also please add a reference for that equation that you used for force calculations.

In this study, the impact force was assessed with a 2D grid size of approximately 10 m. The force is calculated as a resultant force acting on surfaces of buildings in the two horizontal directions. In addition, since the maximum impact force is represented by evaluating the maximum value at each evaluation point regardless of the time. We will add these explanations to the manuscript.

(17)

Figure 10: what is RBF? Mention the full name in the captions.

We will add the full name of RBF (Radial Basis Function) in Figure 10.

(18)

Figure 11: mention the legend of the color bar in the Figure. Is that maximum impact force? Add it to the figure.

The values of the color bars in Figure 11 indicate the impact force and inundation as shown at (a) (b) below the maps. We consider that the legend is not necessary because "(a) impact force" and "(b) inundation" represent the explanation of the color bars. We ask for your understanding.

(19)

Figure 13: what is colour bar? Mention it in the figure.

It indicates exceedance probability. We will add the label near the color bar of Figure 13.

---

## Author Comment (AC2)

**Response to referee comment 2**

The manuscript presents a surrogate-based approach for probabilistic assessment and investigation of the optimal placement of facilities under tsunami forces. The topic is interesting and the manuscript is generally well written, and the presented approach could be used as an efficient alternative for probabilistic risk assessment of coastal infrastructure assets. Therefore, the manuscript is valuable to be published after some revisions.

Thank you for your valuable comments. Our responses are summarized below.

1. In the numerical simulation of the tsunami, results obtained from the 2D analysis are used as the input of the 3D analysis. Please clarify the validity of this simplification and show some comparison results (if any) with theoretical results or experiments.

The validity of the connection between the 2D and 3D analyses was discussed by Takase et al. (2016). Its performance was demonstrated through comparison with the experimental result. We will add this explanation in the revised manuscript.

2. When calculating the tsunami force, what was the exact mesh sensitivity in the full-scale analyses? Did the authors conduct the mesh sensitivity to ensure that the mesh sensitivity is the same for the whole range of the investigated random parameters?

Thank you for pointing out the important discussion point. We have roughly checked mesh sensitivity by performing the simulations with different mesh sizes. Then we determined the mesh size with sufficient resolution to represent the actual inundation depth observed in the Tohoku tsunami. Nevertheless, it is hard to claim that we have adequately examined the reliability of our fluid force calculations. Since we have concentrated on the development of the presented framework and paid scant attention to the mesh sensitivity of the force acting on the building considering the whole range of the investigated random parameters. When applying the proposed method to actual problems, we will examine in detail the reliability of our numerical simulations in the future. We ask for your understanding.

3. According to the results shown in Figure 12, most of the prediction errors are larger than 10%, indicating that the samples may not sufficiently large enough for the surrogate model to be generalized well. How to determine the appropriate number of samples is an important ingredient in data-driven approaches. I suggest the author should define (or add as the future research work) an appropriate stopping criterion to determine whether the sample size is sufficient or not, or resort to the so-called adaptive surrogate modelling to reach a balanced performance by sequentially adding new samples to the training set.

Thank you for your suggestion. As you pointed out, improving the accuracy of surrogate model and deciding an appropriate number of samples for an accurate surrogate model are important and common challenges for data-driven models. Because we used the simulation data set which has been already published, we cannot discuss the appropriate number of samples in this paper. However, since the point the reviewer pointed out is really important, we will add some discussion in the revised manuscript and summarize the point as one of the future works.

4. The manuscript is well written, yet the language should be double-checked to make it more readable, e.g., "Although different placements were obtained, the risk for both parallel and series systems as shown in Table 3" on Page 21.

Thank you for pointing this out. The sentence you pointed out was incorrect. We will carefully check the entire manuscript by all co-authors and revise the manuscript.

---

## Author Response (AR1)

**Responses to review comments**

*We deeply appreciate the reviewers' valuable comments and kind suggestions. We have revised the manuscript in line with the review comments. Responses to reviewers' comments and revision details are summarized below. The revised parts are made in red in the following responses and the revised manuscript.*

**Editor comment**

Following the advice from our referees, who both agreed that your results are interesting and worth to be published, and after a careful check of your response to their comments, I am delighted to invite you to submit your revised manuscript for further consideration and possible publication in our journal after moderate/major revision. First, however, I will need you to revise your article by implementing all the points raised by the reviewers. Particularly, make sure you better clarify the numerical simulation methods and results, improve your figures (that sometime are similar to those from previous works), and add some recent citations on PTHA (there are many in this field) and Tonga global tsunami that can be easily found.

*Thank you for your comments. We have answered all the points raised by the reviewers and revised the manuscript accordingly. We also have tried to better clarify the results, the simulation methods, figures, and added some citations on PTHA and Tonga global tsunami than before. Our revisions in the manuscript are summarized below.*

**Referee comment 1**

Evaluation:

This is a good work and I think can be published. But the text is unclear in some places and misses important citations. It is important that authors clarify the items mentioned below, and add the suggested citations in order to make their work more international. My comments are given below. I look forward to reviewing the revised version. Best regards.

*Thank you for your valuable comments. Our responses are summarized below.*

[1-1]

Abstract: Abstract is too long and is non-conclusive. Please try to add more quantitative results in your abstract. Also I suggest making your abstract shorter.

*Thank you for your kind suggestion. We have revised the abstract to make it shorter and to make the conclusions clearer.*

[Original manuscript, Page 1, Line 1-21, Abstract]

Tsunamis are associated with numerous uncertainties. Therefore, there has been an emphasis on setting the placement of infrastructure facilities based on probabilistic approaches. However, advanced numerical simulations have been often insufficiently utilized due to high computational costs. Therefore, in this study, we developed a framework that could efficiently utilize the information obtained from advanced numerical simulations for probabilistic assessment and investigation of the optimal placement of facilities based on calculated probability. Proper orthogonal decomposition (POD) techniques were employed for utilizing the data from the numerical simulations for probabilistic risk evaluation. We constructed a surrogate model in which POD was efficiently used to extract the spatial modes. The results of the numerical simulation were expressed as a linear combination of the modes, and the POD coefficients were expressed as a function of the uncertainty parameters to represent a result of an arbitrary scenario at a low computational cost. We conducted numerical simulations of the 2011 tsunami off the Pacific Coast caused by Tohoku Earthquake as an example of the method proposed in this study. The tsunami reached the target area, and the fault parameters of "slip" and "rake" were selected as the target uncertainties. We then created several scenarios in which these parameters were changed and conducted further numerical simulations using POD to construct a surrogate model. We selected the maximum inundation depth in the target area and the maximum impact force that acts on the buildings as the target risk indices, and we constructed a surrogate model of the spatial distributions of each indicator. Furthermore, we conducted Monte Carlo simulations using the constructed surrogate model and the information on fluctuations in uncertainties to calculate the spatial distribution of the failure criterion exceedance probabilities. We then used the Monte Carlo simulation results and a genetic algorithm to identify the optimal placement of facilities based on probability. We also discuss how the optimal placement changes according to differences in risk indices and the differences between parallel and series systems. The failure scenarios for each system are also discussed based on the failure probability. We show that the proposed method of efficiently utilizing advanced numerical simulation information was useful for conducting probabilistic hazard assessments and investigating the optimal placement of facilities based on

probability theory.

[Revised manuscript, Page 1, Line 1-10, Abstract]

This study proposes a framework that can efficiently utilize the information obtained from advanced tsunami simulation for probabilistic tsunami hazard assessment (PTHA) and investigation of optimal facility placement. A series of numerical simulations of the tsunami off the Pacific Coast caused by the 2011 Tohoku Earthquake is performed considering uncertainties of fault parameters. The simulated tsunami force acting on buildings and inundation depth are calculated in the simulations, and they are defined as tsunami hazard indices. Proper orthogonal decomposition (POD) is then applied to the simulated results to extract the characteristic spatial modes, which can be used to construct surrogate models. Monte Carlo simulations (MCS) are then performed at a low computational cost using the surrogate models. Based on the MCS results along with the concept of system failure probability, the optimal placement of facilities is probabilistically investigated with the help of genetic algorithms. The results indicate that the proposed framework enables us to determine the optimal placement of facilities applying different strategies at low computational cost while effectively reflecting the results of advanced tsunami simulations.

[1-2] Figure is very similar to author team another article (https://doi.org/10.5194/nhess-22-1267-2022). Please try to make this figure different from your already-published article.

*Thank you for your kind comments. As the reviewer pointed out, some figures in the manuscript are the same as those in the previously published paper(https://doi.org/10.5194/nhess-22-1267-2022). The reason is that this study reuses the tsunami simulation results reported in the paper. We should have written about this fact. We have added the information about the citation in the figures' captions.*

[Original manuscript, Page 10, Table 1]

Names of the calculation cases.

[Revised manuscript, Page 10, Table 1]

Names of the calculation cases. (Tozato et al. (2022))

[Original manuscript, Page 11, Figure 4]

Boundary between 2D and 3D analyses areas. Points A to H are used to compare the inundation depths between the observational data and simulation results. (©Google Maps)

[Revised manuscript, Page 11, Figure 4]

Boundary between 2D and 3D analyses areas. Points A to H are used to compare the inundation depths between the observational data and simulation results. (©Google Maps, Tozato et al. (2022))

[Original manuscript, Page 12, Figure 5]

Snapshots of the tsunami runup obtained through 3D analysis.

[Revised manuscript, Page 12, Figure 5]

Snapshots of the tsunami runup obtained through 3D analysis. (Tozato et al. (2022))

[Original manuscript, Page 12, Figure 6]

Comparison of the inundation between observational data and numerical simulation. (Observation data are provided by field survey results (The 2011 Tohoku Earthquake Tsunami Joint Survey Group, 2012)).

[Revised manuscript, Page 12, Figure 6]

Comparison of the inundation between observational data and numerical simulation. (Observation data are provided by field survey results (The 2011 Tohoku Earthquake Tsunami Joint Survey Group, 2012)). (Tozato et al. (2022))

[Original manuscript, Page 13, Figure 7]

Image of a mesh for evaluating tsunami force.

[Revised manuscript, Page 13, Figure 7]

Image of a mesh for evaluating tsunami force. (Tozato et al. (2022))

[Original manuscript, Page 24, Figure A1]

Configuration of the test structure.

[Revised manuscript, Page 24, Figure A1]

Configuration of the test structure. (borrowed from Winter et al. (2020))

[1-3]

L26: what is meant by "…are not very compatible with probabilistic…?" please clarify.

*We apologize for the incomplete sentences. As the reviewer pointed out, the meaning of the sentence is unclear. We have revised the sentence as follows:*

[Original manuscript, Page 2, Line 25-27]

However, advanced numerical analyses are generally computationally expensive. Furthermore, they are not very compatible with probabilistic assessments and require multiple trials. Therefore, a framework that can efficiently combine these aspects is required.

[Revised manuscript, Page 1, Line 17-19]

However, advanced numerical analyses are unsuitable for probabilistic hazard assessment because a large number of calculation cases are generally required. Therefore, it is necessary to develop a framework that effectively use the few but reliable numerical simulation results to achieve probabilistic evaluations.
* * *
[1-4]

L33/34: For PTHA, to give more diversity to your citations, please add the following two good articles: "Gopinathan et al. 2021" and "Heidarzadeh & Kijko 2011".

Gopinathan, D., Hidarzadeh, M., Guillas, S. (2021). Probabilistic Quantification of tsunami current hazard using statistical emulation. Philosophical Transactions of the Royal Society A, 477, 20210180. https://doi.org/10.1098/rspa.2021.0180.

Hidarzadeh, M., Kijko, A. (2011).  A probabilistic tsunami hazard assessment for the Makran subduction zone at the northwestern Indian Ocean. Natural Hazards, 56 (3), 577-593. https://doi.org/10.1007/s11069-010-9574-x.

L68: For the 2011 event please add a reference for clarity. I recommend the article by Prof Tsuji (Tsuji et al., 2011), as below:

Tsuji Y, Satake K, Ishibe T, Kusumoto S, Harada T, Nishiyama A, Kim HY, Ueno T, Murotani S, Oki S, Sugimoto M, Tomari J, Heidarzadeh M, Watada S, Imai K, Choi BH, Yoon SB, Bae JS, Kim KO, Kim HW (2011) Field surveys of tsunami heights from the 2011 Off the Pacific Coast of Tohoku, Japan, earthquake. Bulletin of the Earthquake Research Institute University of Tokyo 86:29–279 (in Japanese with English abstract)

L72: I think it is useful to add a line regarding the importance of this study. I recommend something like this: "The recent 2022 Tonga tsunami, which made global impacts (Heidarzadeh et al. 2022), showed that tsunami is an important costal disaster and studies like this work are needed".

The reference is:

Heidarzadeh, M., Gusman, A., Ishibe, T., Sabeti, R., Šepić, J. (2022). Estimating the eruption-induced water displacement source of the 15 January 2022 Tonga volcanic tsunami from tsunami spectra and numerical modelling. Ocean Engineering, 261, 112165. https://doi.org/10.1016/j.oceaneng.2022.112165.

*Thank you for the list of suggested references. We have cited some additional references related to PTHA, the 2011 Tohoku tsunami event, and the 2022 Tonga tsunami event and added relevant descriptions in the revised manuscript as necessary.*

**Probabilistic Tsunami Hazard Assessment (PTHA)**

[Original manuscript, Page 2, Line 30-35]

Furthermore, probabilistic tsunami hazard analyses (PTHA) have been established based on PSHAs; they have been proposed as a method for understanding the relationship between tsunami heights and exceedance probabilities in a specified period. PTHAs are practical methods for mitigating disaster damage, and many studies on PTHA have been reported (e.g., Annaka et al., 2007; Power et al., 2007; Thio et al., 2007; Mitsoudis et al., 2012; Fukutani et al., 2021, 2015; Park and Cox, 2016); many review papers have also reviewed the topic (Geist and Parsons, 2006; Mori et al., 2017).

[Revised manuscript, Page 2, Line 22-34]

Furthermore, probabilistic tsunami hazard analyses (PTHA) have been established based on PSHAs; they have been proposed as a method for understanding the relationship between tsunami heights and exceedance probabilities in a specified period ; see, e.g., Geist and Parsons (2006); Annaka et al. (2007); Fukutani et al. (2015); Mori et al. (2017). Also, some studies have focused on actual regions such as the southeast Aegean Sea (Mitsoudis et al., 2012), the Makran subduction zone (Hidarzadeh and Kinjo, 2011), the Cascadia subduction zone (Park and Cox, 2016; LeVeque et al., 2016; Salmanidou et al., 2021) and Nankai Trough subduction zone (Nakano et al., 2020; Baba et al., 2022). In addition, the slip distributions have been investigated based on probabilistic approaches; see, e.g., LeVeque et al. (2016); Nakano et al. (2020); Scala et al. (2020). Moreover, numerous studies have been conducted to efficiently utilize numerical simulation by constructing surrogate models and utilizing them in PTHA. For instance, response surface-based approaches using polynomial functions (e.g., Kotani et al., 2020), the radial basis function (e.g., Gopinathan et al., 2021), and the Gaussian process regression (e.g., Salmanidou et al., 2021; Alhamid et al., 2022) have been reported. A surrogate model constructed based on the concept of the singular value decomposition is worthy of remark because spatial modes are efficiently utilized; see Fukutani et al. (2021).

[Additional references]

[revised manuscript text omitted]

Figure 1: the color bar cannot be read. Please ensure they can be read easily and add a legend for them. Are they "Wave amplitude (m)?", add them.
* * *
*Thank you for pointing it out. Since spatial distributions shown in the Fig. 1 are just images, and color bars are not important. To avoid unnecessary confusion, the color bars have been removed and the*

*images included in Figure 1 have been modified.*

[Original manuscript, Page 4, Figure 1]

[Revised manuscript, Page 4, Figure 1]

[Figure]

[Figure]

[1-6]

L112: The 3D model is not clear? Please add a few references for that and explain more about it.

*Thank you for your kind suggestion. Because the explanation of the 3D model is written in detail in the previous paper (Tozato et al. (2022)), we have added a sentence that directs readers to that paper in the revised manuscript as follows:*

[Original manuscript, Page 5, Line 112]

To solve the governing equations of the 3D simulation, the stabilized finite element method (SFEM) is used in this study.

[Revised manuscript, Page 5, Line 111-112]

To solve the governing equations of the 3D simulation, the stabilized finite element method (SFEM) is used in this study. The detailed explanation of the numerical method used in this study is described in the relevant paper (Tozato et al., 2022).
* * *
[1-7]

L113: What is this boundary condition? Explain more about it by adding the boundary equation and a few references.
* * *
*The explanation of the boundary condition of the 2D and 3D analysis is explained in lines 98-101 on page 5. The tsunami wave height and flow velocity obtained by 2D simulation are set as the boundary conditions for 3D simulation, and the tsunami run-up reaching the target area is simulated in 3D analysis. Although the reference paper was cited in the original manuscript, we understood that the information was not well positioned. We have changed the position as below.*

[Original manuscript, Page 5, Line 112-113]

The method in Takase et al. (2016) was used for the boundary conditions of the 2D and 3D analyses.

[Revised manuscript (Deletion)]

[Original manuscript, Page 5, Line 98-101]

The obtained tsunami wave height and flow velocity were set as the boundary conditions, and the tsunami reaching the target area was analyzed. The time-series data of the wave height and flow velocity obtained from the 2D wide-area analysis are stored and transferred to the 3D numerical analysis by linear interpolation in space. The interpolated values are given to the 3D analysis as input data.

[Revised manuscript, Page 5, Line 97-100]

The obtained tsunami wave height and flow velocity were set as the boundary conditions, and the tsunami reaching the target area was analyzed. The method proposed by Takase et al. (2016) is used for the boundary conditions of the 2D and 3D analyses. The time-series data of the wave height and flow velocity obtained from the 2D wide-area analysis are stored and transferred to the 3D numerical analysis by linear interpolation in space. The interpolated values are given to the 3D analysis as input data.

[1-8]

L120: What are these data? Sea level? Velocity? Force?

*The "these" data indicate either the tsunami force or the inundation depth. To make them clear, we have added an explanation in the revised manuscript as follows:*

[Original manuscript, Page 6, Line 117-122]

Proper orthogonal decomposition (POD) was used to extract the spatial modes in this study. POD can efficiently express data and extract the basis representing data characteristics. To apply POD to the data obtained from numerical analysis, a data matrix was first defined. When data from a given scenario $i$ are set as a vector and defined as $\boldsymbol{x}_i$ (called a data vector), then the result of the data vectors corresponding to the number of scenarios arranged in the column direction is defined as follows:

$$X = \begin{pmatrix} | & & | \\ \boldsymbol{x}_1 & \cdots & \boldsymbol{x}_N \\ | & & | \end{pmatrix} \tag{1}$$

Here, $N$ refers to the number of scenarios, and the data vector is defined as a vector with a total of $n$ elements.

[Revised manuscript, Page 6, Line 117-124]

In this study, proper orthogonal decomposition (POD) was used to extract the spatial characteristic modes from the results of numerical analysis. To apply POD, a column vector $\boldsymbol{x}_i$ (called a data vector) was first defined to accommodate a tsunami hazard index, for which we selected the spatial distribution of either the maximum impact force acting on the buildings or the maximum inundation depth obtained from a numerical simulation for scenario *i*. Here, if there are *n* evaluation points, $\boldsymbol{x}_i$ has *n* components. Then, a data matrix was created by arranging all the data vectors according to a certain rule to be used for a target of POD.

$$X = \begin{pmatrix} | & & | \\ \boldsymbol{x}_1 & \cdots & \boldsymbol{x}_N \\ | & & | \end{pmatrix} \tag{1}$$

where $N$ refers to the number of scenarios.

[1-9]

L131: what is Lamda_k? what is d_j? every parameter needs to be defined as soon as they are used.

*Thank you for pointing it out. $d_j$ represents the contribution rate for mode $j$, and $\lambda_k$ represents the k-th eigenvalue. The index $k$ is used as an index for summation. Since the explanations of $d_j$ and*

*index $k$ were not written in the original manuscript, we have added them in the revised manuscript.*

[Original manuscript, Page 6, Line 128-130]

The contribution rate is an index that shows how much each mode explains the data, and the contribution rate of the $j$-th mode is expressed as follows, using the eigenvalues.

$$d_j = \frac{\lambda_j}{\sum_{k=1}^{n} \lambda_k} \qquad (9)$$

[Revised manuscript, Page 6, Line 129-131]

The contribution rate is an index that shows how much each mode explains the data, and the contribution rate of the $j$-th eigenmode $d_j$ is expressed as follows, using the eigenvalues $\lambda_k$ ($k = 1, ..., n$)

$$d_j = \frac{\lambda_j}{\sum_{k=1}^{n} \lambda_k} \qquad (9)$$

[1-10]

L139: again some parameters are not defined. Please ensure you define al parameters as soon as they are used throughout the text.

*Thank you for pointing this out. Since it was not clearly mentioned that $\alpha_{ik}$ is a component of matrix **A**, we have added this explanation in the revised manuscript.*

[Original manuscript, Page 6, Line 139]

Here, $\alpha_{ik}$ shows the $k$-by-$i$ column component of the matrix in which the POD coefficients are arranged.

[Revised manuscript, Page 6, Line 140]

Here, $\alpha_{ik}$ denotes *ik* component of the matrix **A** in which the POD coefficients are arranged.

[1-11]

L200-204: it is not clear why authors considered only slip and rake for uncertainty? Please clarify this and try to convince the readers? Why not depth while depth is a very important factor regarding tsunami energy? Please clarify.

*Thank you for checking the details. The reason why this study only considers slip and rake as uncertainty parameters is that these parameters are supposed to have a deep relationship with the*

*characteristics of fault stagger. As mentioned in Response to Comment 1-2, this study reuses the simulation data that were provided by our previous work (Tozato et al., 2022), and the reason why we employed the two fault parameters are written in that paper. Since this study mainly aims to propose a framework for the probabilistic optimal facilities placement, we thought that it is better to refer to the previous paper for the details of the tsunami simulation and to provide the minimum description in this study. However, as you suggested, an additional explanation in this work might help the readers' better understanding. So, we have added an explanation about the slip and the rake in the revised manuscript as follows:*

[Original manuscript, Page 9, Line 200-203]

We used the same numerical analysis results as those conducted in previous research (Tozato et al., 2022). The reader is referred to the study by (Tozato et al., 2022) for details regarding the computational conditions. The parameter values at the time of the Off the Coast of Tohoku Earthquake were set as mean values, with the slip varying between 0.7 and 1.4 times, and the rake varying between −20◦ to +25◦.

[Revised manuscript, Page 9, Line 200-204]

We used the same numerical analysis results as those conducted in previous research (Tozato et al., 2022). The reader is referred to the study by (Tozato et al., 2022) for details regarding the computational conditions. This study only considers the slip and rake as uncertainty parameters shown in Fig. 3 because these parameters are supposed to have a deep relationship with the characteristics of fault stagger. The parameter values at the time of the Off the Coast of Tohoku Earthquake were set as mean values, with the slip varying between 0.7 and 1.4 times, and the rake varying between −20◦ to +25◦.
* * *
[1-12]

L203: regarding rake, you could refer to the following good articles that studied rake variations through teleseismic inversions (Gusman et al., 2014; Heidarzadeh et al., 2017):

Gusman, A. R., Murotani, S., Satake, K., Heidarzadeh, M., Gunawan, E., Watada, S., & Schurr, B. (2015). Fault slip distribution of the 2014 Iquique, Chile, earthquake estimated from ocean-wide tsunami waveforms and GPS data. Geophysical Research Letters, 42, 1053-1060. https://doi.org/10.1002/2014GL062604.

Heidarzadeh, M., Murotani, S., Satake, K., Takagawa, T., Saito, T. (2017). Fault size and depth extent of the Ecuador earthquake (Mw 7.8) of 16 April 2016 from teleseismic and tsunami data. Geophysical

Research Letters, 44 (5), 2211–2219. https://doi.org/10.1002/2017GL072545.

*Thank you for the valuable information. We have checked the articles. We think the main scope of these studies is an inversion of the slip distribution, and the variations of the rake are not taken into account in these studies. It is therefore difficult to refer to the articles as citations of rake variations in the manuscript. We ask for your understanding.*

[1-13]

Figure 5: I assume that the elevation is "Topography elevation (m)". Please modify the legend.

*Thank you for pointing it out. We have revised the legend in Figure 5.*

[Original manuscript, Page 12, Figure 5]

[Figure]

[Revised manuscript, Page 12, Figure 5]

[Figure]

[1-14]

L208: For the 2011 Tohoku data, add reference to Mori et al. (2011):

Mori, N., Takahashi, T., and The 2011 Tohoku Earthquake Tsunami Joint Survey Group, (2012), Nationwide post event survey and analysis of the 2011 Tohoku earthquake tsunami. Coastal Engineering Journal, 54 (1), 1-27. https://doi.org/10.1142/S0578563412500015.

*Thank you for your kind suggestion. We have cited the paper in the manuscript.*

[Original manuscript, Page 9, Line 207-209]

Figure 6 shows the results of comparison of the inundation depth with the actually observed inundation depth. The observed data were referenced from (The 2011 Tohoku Earthquake Tsunami Joint Survey Group, 2012), and the placement of the observation points from A to H is shown in Fig. 4.

[Revised manuscript, Page 9-11, Line 209-212]

To confirm the validity of the numerical simulations, the simulation result for case S3R5, which corresponds to the actual tsunami condition, is compared with the observed data (The 2011 Tohoku Earthquake Tsunami Joint Survey Group, 2012; Mori et al., 2012). Figure 6 shows the simulated and observed inundation depths at the points A to H indicated in Fig. 4.

[Additional references]

Mori. N., Takahashi, T., THE 2011 Tohoku Earthquake Tsunami Joint Survey Group: Nationwide

Post Event Survey and Analysis of the 2011 Tohoku Earthquake Tsunami, Coastal Engineering Journal, 54(1), 1250001-1-1250001-27, https://doi.org/10.1142/S0578563412500015, 2012.

[1-15]

Figure 8: what is the color bar? Please write it in the figure next to the colour bar. This way the figure can be easier and readers can notice it instantly, instead of reading your caption.

Thank you for your valuable comment. We have added the explanation to the legend of Figure 8.

[Original manuscript, Page 14, Figure 8]

[Figure]

[Revised manuscript, Page 14, Figure 8]

[Figure]

(a) Impact force

(b) Inundation

[1-16]

L254: It is not clear how the maximum impact force is calculated? Please write the equation that you used to calculate the impact force here. Also please add a reference for that equation that you used for force calculations.

*In this study, the impact force was evaluated with a 2D mesh consisting of approximately 10 m x 10 m grids. The impact force is defined as a resultant force (pressure) acting on the surfaces of buildings in the two horizontal directions. In addition, for each grid, the maximum impact force is represented by evaluating the maximum value over the analysis time. We have added the following explanations to the manuscript:*

[Original manuscript, Page 11, Line 219-220]

Therefore, the tsunami fluid force was assessed with a 2D grid size of approximately 10 m in this study. An image of a mesh for evaluating the tsunami force is shown in Fig. 7.

[Revised manuscript, Page 11, Line 221-226]

Therefore, the tsunami fluid force was assessed with a 2D mesh consisting of approximately 10 m x 10 m grids in this study, each of which is a unit for force evaluation. The tsunami impact force is calculated by synthesizing all the pressures acting on the surfaces of buildings within each grid in the two horizontal directions and averaged over the grid. Then, hereafter, each grid is regarded as a point associated with this averaged force. In addition, for each point, the maximum impact force is represented by evaluating the maximum value over the analysis time. An image of a mesh for evaluating the tsunami force is shown in Fig. 7.
* * *
[1-17]

Figure 10: what is RBF? Mention the full name in the captions.
* * *
*We apologize for the lack of information. Although its full name is written in the text, we agree that the additional information could be helpful for readers. We have added the full name of RBF (Radial Basis Function) in Figure 10.*

[Original manuscript, Page 15, Figure 10]

[Figure]

[Revised manuscript, Page 16, Figure 10]

[Figure]

[1-18]

Figure 11: mention the legend of the color bar in the Figure. Is that maximum impact force? Add it to the figure.

*Thank you for your valuable suggestion. As you pointed out, we have added the legend of the color bars in Figure 11.*

[Original manuscript, Page 16, Figure 11]

[Figure]

(a) Impact force

(b) Inundation

[Revised manuscript, Page17, Figure 11]

[Figure]

(a) Maximum impact force

(b) Maximum inundation depth

[1-19]

Figure 13: what is colour bar? Mention it in the figure.

*Thank you for pointing it out. It indicates the exceedance probability. We have added the label near the color bar in Figure 13.*

[Original manuscript, Page 18, Figure 13]

[Figure]

(a) Impact force       (b) Inundation

[Revised manuscript, Page 19, Figure 13]

[Figure]

(a) Impact force       (b) Inundation

**Referee comment 2**

> The manuscript presents a surrogate-based approach for probabilistic assessment and investigation of the optimal placement of facilities under tsunami forces. The topic is interesting and the manuscript is generally well written, and the presented approach could be used as an efficient alternative for probabilistic risk assessment of coastal infrastructure assets. Therefore, the manuscript is valuable to be published after some revisions.

*Thank you for your valuable comments. Our responses are summarized below.*

> [2-1]
> In the numerical simulation of the tsunami, results obtained from the 2D analysis are used as the input of the 3D analysis. Please clarify the validity of this simplification and show some comparison results (if any) with theoretical results or experiments.

*The validity of the connection between the 2D and 3D analyses was discussed by Takase et al. (2016) and its performance was demonstrated through comparison with the experimental result. In addition, in the present study, the validation is also confirmed by comparing the numerical results with the observed inundation depth as shown in Fig. 6. We have added the explanation in the revised manuscript.*

[Original manuscript, Page 9, Line 207-209]
Figure 6 shows the results of comparison of the inundation depth with the actually observed inundation depth. The observed data were referenced from (The 2011 Tohoku Earthquake Tsunami Joint Survey Group, 2012), and the placement of the observation points from A to H is shown in Fig. 4.

[Revised manuscript, Page 9-11, Line 209-212]
To confirm the validity of the numerical simulations, the simulation result for case S3R5, which corresponds to the actual tsunami condition, is compared with the observed data (The 2011 Tohoku Earthquake Tsunami Joint Survey Group, 2012; Mori et al., 2012). Figure 6 shows the simulated and observed inundation depths at the points A to H indicated in Fig. 4.

> [2-2]
> When calculating the tsunami force, what was the exact mesh sensitivity in the full-scale analyses?

Did the authors conduct the mesh sensitivity to ensure that the mesh sensitivity is the same for the whole range of the investigated random parameters?

*Thank you for providing the important discussion point. As mentioned in the main body and Responses to Comments 1-2 and 1-11, the simulation results obtained in our previous work is reused in this study. In the previous work, we have roughly checked mesh sensitivity by performing the simulations with different mesh sizes. Then we determined an adequate mesh size to calculate the inundation depth comparable to the actually observed one in the 2011 Tohoku tsunami. Although it is hard to claim that we have thoroughly examined the mesh dependency, we believe that the simulation results have a certain degree of reliability. We ask for your understanding.*

[2-3]

According to the results shown in Figure 12, most of the prediction errors are larger than 10%, indicating that the samples may not sufficiently large enough for the surrogate model to be generalized well. How to determine the appropriate number of samples is an important ingredient in data-driven approaches. I suggest the author should define (or add as the future research work) an appropriate stopping criterion to determine whether the sample size is sufficient or not, or resort to the so-called adaptive surrogate modelling to reach a balanced performance by sequentially adding new samples to the training set.

*Thank you for your important remarks and kind suggestion. As you pointed out, improving the accuracy of surrogate model and deciding an appropriate number of samples are important to obtain a reliable surrogate model. Indeed, these are common challenges for data-driven modeling. Because we used the simulation data set which has been already published, it is difficult to discuss the appropriate number of samples in this paper. However, since the reviewer's point is very important, we have added some discussion in the revised manuscript and left it to future work.*

[Original manuscript, Page 16, Line 262-267]

Figure 11 shows a comparison of the numerical simulation results and the results obtained from the surrogate model for the S3R3 scenario. Regarding the number of modes used in the surrogate model, the maximum impact force was set to 8 and the maximum inundation depth was set to 11. The validity of the surrogate model can be confirmed because the figure shows that the spatial distribution is generally reproduced. Furthermore, Fig. 12 shows the results of assessing the errors of the 10 scenarios that were held for verification. The error was calculated using Eq. (18). The error values as well that the numerical analysis results were generally reproduced.

[Revised manuscript, Page 15-16, Line 266-273]

Figure 11 shows a comparison between the numerical simulation results and the surrogate model results for the S3R3 scenario. Regarding the number of modes used in the surrogate model, the maximum impact force was set to 8 and the maximum inundation depth was set to 11. Also, Fig. 12 shows the mean absolute errors calculated by Eq. (18) in calculations for 10 scenarios used for validation. As can be seen from Fig. 11, the constructed surrogate models can roughly represent the targeted spatial distribution of the maximum impact force and maximum inundation depth. However, Fig. 12 shows that 10 % or higher errors occur in some validation scenarios, and the areas of large error are localized. This is because there is a possibility that spatial modes used for the surrogate models cannot properly capture the local characteristics.

[Original manuscript, Page 23, Line 367-368]

In addition, since the accuracy of the surrogate model changes according to the number of spatial modes, it's necessary to establish a way of properly determining the number of modes in future works.

[Revised manuscript, Page 24, Line 374-380]

In addition, since the accuracy of the surrogate model changes according to both the number of training data and the number of spatial modes used in the surrogate model, it is necessary to establish a way of properly determining them in future studies. In this study, we used the simulation data set that was created in the previous study and therefore could not investigate their effect for surrogate modeling. Nevertheless, an appropriate number of training data should be carefully determined in light of accuracy. To address this problem, it may be necessary to consider adopting an approach like Adaptive Surrogate Modeling (e.g., Wang et al., 2014; Gong et al., 2015) to check and improve the accuracy of surrogate models.

[Additional references]

Gong, W., Duan, Q., Li, J., Wang, C., Di, Z., Ye, A., Miao, C., and Dai, Y.: Multiobjective adaptive surrogate modeling-based optimization for parameter estimation of large, complex geophysical models, Water Resour. Res., 52, 1984– 2008, https://doi.org/10.1002/2015WR018230, 2016.

Wang, C., Duan, Q., Gong, W., Ye, A., Di, Z., Miao, C.: An evaluation of adaptive surrogate modeling based optimization with two benchmark problems, Environmental Modelling & Software, 60, 167-179, https://doi.org/10.1016/j.envsoft.2014.05.026, 2014.

[2-4] The manuscript is well written, yet the language should be double-checked to make it more readable, e.g., "Although different placements were obtained, the risk for both parallel and series systems as shown in Table 3" on Page 21.

*Thank you for pointing this out. Indeed, the sentence you pointed out was unclear. We have revised this sentence and have carefully double-checked the entire manuscript to make it more readable.*

[Original manuscript, Page 21, Line331-333]

Through the comparison of the obtained optimal placements for all systems, it can be seen that some common locations were selected for some components, but different locations were also selected for others. Although different placements were obtained, the risk for both parallel and series systems as shown in Table 3.

[Revised manuscript, Page 20, Line 337-340]

The comparison between the placement selected by the genetic algorithms and the placement selected in order of lower exceedance probabilities shows that some selected locations are different from each other. The system failure probabilities shown in Table 3 indicate that lower failure probability tends to be obtained by the genetic algorithm. Thus, it can be confirmed that the genetic algorithm is suitable in this particular optimal placement problem.

**Other revisions**

*We have revised the sentence and have carefully checked the entire manuscript. In addition, we have changed the color schemes of the maps in Fig. 1, 8, 11,13, and 14 to make all readers correctly interpret.*

[Original manuscript, Page 20, Figure 14]

[Figure]

[Revised manuscript, Page 21, Figure 14]

---

## Author Response (AR2)

**Response sheet**

We deeply appreciate your valuable suggestions for our manuscript. We have revised it to address them, and the revised parts are colored in red in the revised manuscript. To improve the overall readability and language quality, we utilized a native speaker English proofreading service. The English proofreading certificate is attached. In addition, we have updated the affiliation information because the affiliation of one of the authors has changed.

[Figure]

**Editing Certificate**

This document certifies that the manuscript listed below has been edited to ensure language and grammar accuracy and is error free in these aspects. The logical presentation of ideas and the structure of the paper were also checked during the editing process. The edit was performed by professional editors at Editage, a division of Cactus Communications. The author's core research ideas were not altered in any way during the editing process. The quality of the edit has been guaranteed, with the assumption that our suggested changes have been accepted and the text has not been further altered without the knowledge of our editors.

**MANUSCRIPT TITLE**

**Optimal probabilistic placement of facilities using a surrogate model for 3D tsunami simulations**

**AUTHORS**

**Kenta Tozato**

**ISSUED ON**

**April 20, 2023**

**JOB CODE**

**UJRIG_1_2**

[Figure]

[Figure]

**Vikas Narang**
**Chief Operating Officer - Editage**

**editage**

Editage, a brand of Cactus Communications, offers professional English language editing and publication support services to authors engaged in over 1300 areas of research. Through its community of experienced editors, which includes doctors, engineers, published scientists, and researchers with peer review experience, Editage has successfully helped authors get published in internationally reputed journals. Authors who work with Editage are guaranteed excellent language quality and timely delivery.

**GLOBAL :**
+1(833) 979-0061 | request@editage.com

**JAPAN :**
0120-50-2987 | submissions@editage.com

**CACTUS**

IMPACT SCIENCE  impact.science

researcher.life

CACTUS  lifesciences.cactusglobal.com